# Hydrophobic and Corrosion Behavior of Sol-Gel Hybrid Coatings Based on the Combination of TiO$_2$ NPs and Fluorinated Chains for Aluminum Alloys Protection

**Pedro J. Rivero** [1,2,*]**, Juan Deyo Maeztu** [1,3]**, Calos Berlanga** [1,2]**, Adrian Miguel** [4]**, José F. Palacio** [4] **and Rafael Rodriguez** [1,2] 

[1] Materials Engineering Laboratory, Department of Engineering, Public University of Navarre, Campus Arrosadía S/N, 31006 Pamplona, Spain; deyo.maeztu@nadetech.com (J.D.M.); carlos.berlanga@unavarra.es (C.B.); rafael.rodriguez@unavarra.es (R.R.)

[2] Institute for Advanced Materials (InaMat), Public University of Navarre, Campus Arrosadía S/N, 31006 Pamplona, Spain

[3] Nadetech Instruments S.A. Nave B3, Plaza Cein, 5, 31110 Navarra, Spain

[4] Centre of Advanced Surface Engineering, AIN, 31191 Cordovilla, Spain; amiguel@ain.es (A.M.); jfpalacio@ain.es (J.F.P.)

[*] Correspondence: pedrojose.rivero@unavarra.es; Tel.: +34-948-168961

**Abstract:** In this work, layers of a sol-gel hybrid matrix doped with metal oxide nanoparticles (TiO$_2$ NPs) have been deposited on flat samples of AA6061-T6 aluminum alloy using the dip-coating technique, with the aim of obtaining coatings with better anti-corrosive and hydrophobic properties. Two different organic modified silica alkoxides, namely 3-(glycidyloxypropyl)trimethoxysilane (GPTMS) and methyltriethoxysilane (MTEOS), have been used for an adequate entrapment of the metal oxide nanoparticles. In addition, a fluorinated metal-alkoxide precursor has also been added to the hybrid matrix in order to improve the hydrophobic behavior. The experimental results corroborate that the presence of these TiO$_2$ NPs play an important role in the development of the sol-gel hybrid coatings. The water contact angle (WCA) measurements, as well as pencil hardness tests indicate that TiO$_2$ NPs make a considerable increase in the resultant hydrophobicity possible, with better mechanical properties of the coatings. The coating thickness has been measured by cross-section scanning electron microscopy (SEM). In addition, a glow discharge optical emission spectroscopy (GD-OES) analysis has been carried out in order to corroborate the adequate entrapment of the TiO$_2$ NPs into the sol-gel coatings. Finally, potentiodynamic polarization tests and electrochemical impedance spectroscopy (EIS) have been performed in order to evaluate the corrosion resistance of the coatings. All the results provide insights into the efficacy of the developed sol-gel hybrid coatings for anticorrosive purposes with good mechanical properties.

**Keywords:** sol-gel hybrid matrix; TiO$_2$ NPs; corrosion resistance; hydrophobicity

## 1. Introduction

Aluminum and its alloys, after steel and cast irons, are the group of metallic materials most widely used in industry. Their unique combination of low density, mechanical properties, high thermal and electrical conductivity, and affordable cost make aluminum the first choice material for an increasing number of applications, such as airframes, automotive components, power lines, heat exchangers, or food containers.

Despite the low strength and high ductility of the pure metal, many of its alloys, such as those of the 2XXX, 6XXX, and 7XXX series, can reach excellent mechanical properties through solution and ageing heat treatments [1]. AA6061 is a paradigmatic example, being one of the most employed alloys in aeronautics, as well as in the automobile industry and in shipbuilding. In addition to the above mentioned applications, aluminum is also playing a major role in the development of new technologies for energy production and storage. Bipolar plates employed in Proton exchange membrane (PEM) fuel cells are perhaps the best studied example of aluminum based components working in a chemically aggressive environment [2,3]. On the other hand, among energy storage systems where aluminum electrodes are a key element, it is worth mentioning capacitors [4] and new batteries [5–7]. In all these applications, the prevention of corrosion is a main concern, and multiple investigations are focused on the design of optimum coatings to provide long-lasting corrosion and abrasion protection [8,9]. In this respect, coatings and different surface modification techniques play an important role by providing a means to enhance surface anti-corrosion without affecting the bulk properties of the aluminum alloy. For structural applications, in environments where chloride ions can be present, the use of chromates as corrosion inhibitors has been the standard solution for providing the required protection in aqueous media. In the case of aluminum alloys, this can be implemented chromic acid anodized and chromate conversion coatings [10,11]. However, in the last two decades, the use of hexavalent chromium compounds has been banned in most industrialized countries, because they are extremely dangerous for the ecosystem and human health [12,13]. Due to these serious drawbacks, the scientific community has been looking for safer and ecofriendly alternatives, with the aim of replacing these toxic chromium compounds [14]. In the case of AA6061 bipolar plates, Polyaniline (PANI) and Polypyrrole (PPy) films [2], as well as Physical Vapor Deposition (PVD) coatings for AA5083 [15], have been suggested as possible solutions for preventing corrosion in PEM fuel cells.

Scalability must always be a main concern in the choice of solutions for these new problems. Sol-gel coating technology is conducive to being scaled by implementing spray coating methods, or in particular cases, roll-to-roll processes. In addition, sol-gel technology is a potential candidate for corrosion protection because it shows interesting features, such as a good barrier effect, the possibility of incorporating inhibitors into the matrix, good adhesion over different metallic substrates, and good compatibility with additional top layers. A large amount of sol-gel research for corrosion protection can be found in the bibliography [16–22], because silicon alkoxide precursors show an adequate balance of reactivity and ease of handling combined with a ready availability [23]. However, it has been demonstrated that an effective anti-corrosion action is totally dependent on multiple parameters, such as silica precursor concentration [24], deposition time [25], drying time [20], thermal treatment [26], number of dips [27], and ageing time, as well the pH of the starting solution [28,29]. Moreover, the use of organic–inorganic hybrid sol-gel coatings is more effective in terms of the corrosion protection of metallic substrates, because it is possible to form dense and thicker films [30,31] in the micrometric scale, without the presence of cracks [32,33]. In addition, hybrid films show a better flexibility to accept corrosion inhibitors of inorganic as well as organic nature [34–39], in comparison with only pure precursors.

One of the methods used for corrosion protection based on sol-gel technology is the design of hydrophobic or super-hydrophobic surfaces, because they are able to show a water and aqueous electrolyte repellence, anti-icing, and anti-fouling [40,41]. In this respect, one of the most used strategies is to change the morphology of the outer surface of the sol-gel coating by using fluorinated silica precursors or water based short chain per-fluoro emulsion [42–45]. However, the major drawbacks of using these types of fluorinated precursors as potential coatings are the resultant thickness, wear, and mechanical resistance [42], which can limit a further industrial application. In order to overcome this, the new trends are based on the use of sol-gel coatings modified with inorganic nanoparticles with the aim of obtaining a better mechanical resistance. In this sense, most of the research lines are focused on the incorporation of silica nanoparticles for obtaining sol-gel superhydrophobic surfaces with enhancement in the resultant hardness [46–49]. In addition, other types of metal oxide nanoparticles,

in particular TiO$_2$ nanoparticles (NPs), are used to improve the anti-corrosive properties, as well as the mechanical resistance of the coatings [50–53]. These nanoparticles are prepared in different forms (anatase or rutile forms), showing unique properties such as non-toxicity, excellent antibacterial activity, good compatibility with various materials, high chemical stability at high temperatures, high photocatalytic activity, photo-stability, and the ability to absorb ultraviolet light. A representative example of this great multifunctionality related to TiO$_2$ NPs can be found in Reference [54], where its specific mechanism action is clearly explained, whereas potential applications of titanium dioxide materials can be found in Reference [55]. According to this, the use of these nanoparticles makes possible the fabrication of advanced coatings with improved corrosion resistance and antibacterial properties due to its intrinsic excellent antimicrobial properties [56,57]. Moreover, other research of the TiO$_2$-based materials can be found in environmental and energy related applications, such as photocatalysis and photovoltaics [58,59].

Table 1 presents a summary of some of the different sol-gel alternatives previously commented on based on the design of effective coatings for corrosion protection as a function of the selected metallic substrate, the precursors used for the fabrication of the sol-gel coating, and the presence or absence of active agents.

**Table 1.** Summary of the fabrication of different sol-gel coatings as a function of the selected substrate, precursors, and the presence or absence of active agents.

| Metallic Substrate | Precursors for the Fabrication of the Sol-Gel Coating | Active Agent | Ref. |
|---|---|---|---|
| AA2024-T3 | Tetramethylortosilicate | - | [17] |
| Carbon steel | Zirconium tetrabutoxide | - | [18] |
| Mild steel | 3-glycidoxypropyltrimethoxysilane (GPTMS) and aminopropylethoxysilane | - | [20] |
| AA5754 | Tetraethylorthosilicate (TEOS) | - | [25] |
| AISI 304 | Tetraethylorthosilicate (TEOS) | - | [27] |
| AISI 304 | Tetraethylorthosilicate (TEOS) and 3-methacryloxypropyltrimethoxysilane (MPS) | - | [28] |
| AA2024-T3 | 3-glycidoxypropyltrimethoxysilane (GPTMS) and titanium organic compounds | - | [32] |
| AA2024-T3 | Tetramethoxysilane (TMOS) and 3-glycidoxypropyltrimethoxysilane (GPTMS) | Organic corrosion inhibitors | [34] |
| AA2024-T3 | Vinyltrimethoxysilane (VTMS) and tetraethylorthosilicate (TEOS) | Ethylenediamine tetra (methylene phosponic acid) | [35] |
| AA3005 | Tetraethylorthosilicate (TEOS) and methyltriethoxysilane (MTES) | Cerium salts | [36] |
| AA2024-T3 | Tetramethoxysilane (TMOS) and 3-glycidoxypropyltrimethoxysilane (GPTMS) | Organic corrosion inhibitors | [39] |
| AA6061T6 | Methyltriethoxysilane (MTEOS), 3-glycidoxypropyltrimethoxysilane (GPTMS), and perfluoroalkylsilane | Graphene oxide | [43] |
| AA2024 | Tetraethylorthosilicate (TEOS) and 3-methoxysilylpropylmethacrylate (TSPM) | TiO$_2$-CeO$_2$ nanoparticles | [50] |

Finally, this work proposes a novel type of sol-gel coating based on the incorporation of inorganic titanium dioxide nanoparticles (TiO$_2$ NPs) into a sol-gel hybrid matrix by using a copolymerization process of two organic modified silica alkoxides precursors, namely methyltriethoxysilane (MTEOS) and 3-(glycidyloxypropyl)trimethoxysilane (GPTMS). Then, a new metal alkoxide precursor which contains fluorinated polymeric chains is added to the previous sol-gel hybrid matrix with the aim of improving the hydrophobic behavior. All the experimental data corroborate that the presence of these metal oxide nanoparticles in the hybrid matrices combined with a hydrophobic precursor has been of great importance and relevance in the design of multifunctional coatings for three main reasons. Firstly, a considerable increase in the resultant hydrophobicity of the sol-gel coatings has been observed in comparison with a sol-gel blank coating without nanoparticles. Secondly, an important improvement of the mechanical properties (pencil hardness tests) has been obtained in the coatings

composed of these metal oxide nanoparticles. Thirdly, an enhancement in the corrosion resistance has also been demonstrated using potentiodynamic polarization tests and electrochemical impedance spectroscopy (EIS). In addition, the simplicity, versatility, and ease of the fabrication process using the dip-coating technique allows the development of effective coatings as a potential replacement for chromate treatments. The combination of the barrier protection effect of silica coating and the improvement in the mechanical properties due to the addition of metal oxide nanoparticles can be used for the design of novel protective coatings against corrosion for industrial applications.

## 2. Experimental Section

### 2.1. Materials

Aluminum alloy (6061-T6) was selected as the reference substrate in order to undertake different sol-gel coatings. All the samples were cut, polished, and cleaned, with final dimensions of 100 mm in length, 25 mm in width, and 2.5 mm in thickness. The chemical reagents were purchased from Sigma–Aldrich (Madrid, Spain), namely methyltriethoxysilane (MTEOS), 3-(glycidyloxypropyl)trimethoxysilane (GPTMS), 1H,1H,2H,2H-perfluorooctyltriethoxysilane (PFAS), titanium dioxide nanoparticles ($TiO_2$ NPs), ethanol (EtOH) (>99.5%), hydrochloric acid (HCl, 37%), and sodium chloride (NaCl, 3.5%). All the chemicals were of analytical reagent grade and all the aqueous solutions were prepared using ultrapure deionized water.

### 2.2. X-Ray Diffraction Analysis

XRD analyses of the $TiO_2$ NPs were made using a BRUKER D8 Discover diffractometer (Bruker, Billerica, MA, USA) fitted with a copper source ($K\alpha$ = 1.5406 Å) under the Bragg Brentano configuration and 2θ scanning between 20° and 80°, with 0.02° and 1 s per step. A 0.1 mm nickel sheet was placed just before the detector in order to eliminate the Cu-Kβ radiation on the diffractograms.

### 2.3. Deposition Process of the Coatings

The different steps for the fabrication of the hybrid sol-gel matrix (MTEOS and GPTMS), as well as the chemical structures of the alkoxide precursors, are summarized in Figure 1. The MTEOS-based sol (sol_1) was prepared by mixing MTEOS, 0.1 M aqueous HCl, and EtOH in a 1:0.007:6.25 molar ratio, with a specific ageing time of 5 days. The GPTMS-based sol (sol_2) was prepared by mixing GPTMS, 0.1 M aqueous HCl, and EtOH in a 1:4:6.25 molar ratio, with an aging time of 1 day. Then the GPTMS-MTEOS hybrid sol was prepared by combining the two separate sols in a desired molar ratio of 3:1. In the following step, $TiO_2$ NPs were added to this hybrid sol-gel solution in order to have a final silane-$TiO_2$ NPs ratio of $10^{-3}$. The resultant mixture of the $TiO_2$ NPs doped sol-gel hybrid matrix was aged for a period of 3 days under ambient conditions. In order to increase the hydrophobic property of the hybrid sol-gel matrix composed of $TiO_2$ NPs, a third metal-alkoxide precursor based on fluorinated chains (PFAS) was also prepared [42–45]. The PFAS-based sol (sol_3) was prepared by mixing PFAS, 0.1 M aqueous HCl, and EtOH in a 1:5:4.25 molar ratio, with an aging time of 1 day. Finally, a ND-R rotatory dip coater provided by Nadetech Inc. (Pamplona, Spain) was used for the fabrication of the coatings with a dip speed (immersion and withdrawal) of 100 mm/min, and an immersion time of 300 s for each specific dip.

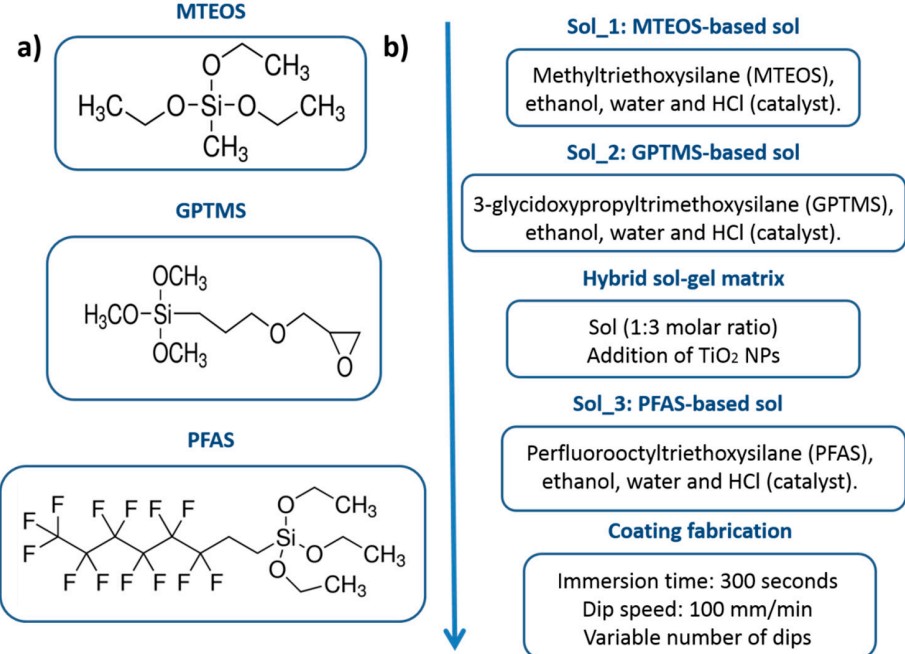

**Figure 1.** (**a**) Chemical structure of the sol-gel metal alkoxide precursors methyltriethoxysilane (MTEOS), 3-(Glycidyloxypropyl)trimethoxysilane (GPTMS), and 1H,1H,2H,2H-Perfluorooctyltriethoxysilane (PFAS). (**b**) Evolution of the different steps involved in the sol-gel process for the fabrication of the $TiO_2$ nanoparticles (NPs) doped sol-gel hybrid coatings with water-repellent behavior.

### 2.4. Characterization of the Sol-Gel Hybrid Coatings

A contact angle meter (CAM 100 KSV Instruments, Burlington, VT, USA) was used for the evaluation of the wettability of all the coatings. The static contact angle was obtained by analyzing the captured images using the tangent method algorithm. An average value of the contact angle between the sample surfaces and a minimum of five deionized water drops was measured using the sessile drop method.

A HITACHI S4800 field emission-scanning electron microscope (FE-SEM, Hitachi S4800, Tokyo, Japan) was used to examine the sample surface and cross section, with the aim of estimating the morphology and thickness of the sol-gel coatings. In addition, in order to corroborate an adequate presence of the different chemical elements in the resultant thickness of the sol-gel hybrid coatings, the samples were analyzed using glow discharge optical emission spectrometry (GD-OES, Horiba, Kyoto, Japan). A JOBIN YVON 10000RF system (Horiba, Kyoto, Japan) was employed and calibrated to quantify the atomic ratio (%) of Ti, F, Si, C, O, and Al. In addition, the topography along with the surface was examined using atomic force microscopy (AFM) in tapping mode (Veeco Innova AFM, Veeco Instruments, Plainview, NY, USA) for a scan area of 20 μm × 20 μm.

The mechanical properties were determined using both the pencil hardness and cross-cut tape tests. The pencil hardness test is based on a constant-load scratch test when pencil leads of different hardness grades from 9B to 9H are applied. The pencil hardness of the coating is obtained when the hardness pencil grade does not cause any damage to the coated sample. In order to determine the adhesion of the coating with the substrate, eleven cuts were made using the cutter in two directions at right angles to each other to form a grid of small squares. Then, a pressure-sensitive adhesive tape was applied over the lattice and removed by pulling in a single smooth action.

### 2.5. Electrochemical Corrosion Tests

All the corrosion tests were conducted through cyclic potentiodynamic polarization techniques and electrochemical impedance spectroscopy (EIS), using an Autolab PGSTAT30 galvanostat/

potentiostat system (Metrohm, Herisau, Switzerland) at room temperature. The experiments were conducted with a three electrode system composed of sol-gel coated aluminum substrate as the working electrode (WE), platinum wire as a counter electrode (CE), and an Ag/AgCl electrode as the reference electrode (RE). It should be noted that before testing, all the samples of the study were exposed for a while (1 h) with the intention of stabilizing the open circuit potential (OCP).

Cyclic potentiodynamic polarization measurements were performed in order to determine the localized corrosion susceptibility. Samples were immersed in saline solution (3.5 wt.% NaCl in ultrapure water) for a period of 1 h before initiating polarization, and then the potential scan started from $E_{corr}$. The scan rate was 0.6 V/h towards the noble direction. Once the current density reached a specific value of 5 mA/cm$^2$, the scan direction was reversed until the hysteresis loop was closed, or until the corrosion potential was reached.

Electrochemical impedance spectroscopy measurements were performed in 3.5 wt.% NaCl solution in a frequency range of 0.01 Hz to 100 kHz, with a wave amplitude of 10 mV at room temperature.

## 3. Results and Discussion

As has been previously commented on in Figure 1, the incorporation of TiO$_2$ NPs into the sol-gel hybrid matrix was performed with the aim of improving the corrosion resistance of the coatings. First of all, in order to clarify the nature of these TiO$_2$ NPs in the sol-gel coatings, an XRD analysis of these nanoparticles was made, and the results are shown in Figure 2. The XRD diffractogram indicates that the resultant titanium oxide used for the fabrication of the sol-gel hybrid coatings consists of a mixture of rutile and anatase, because the presence of both phases can be clearly observed in the XRD diffractogram.

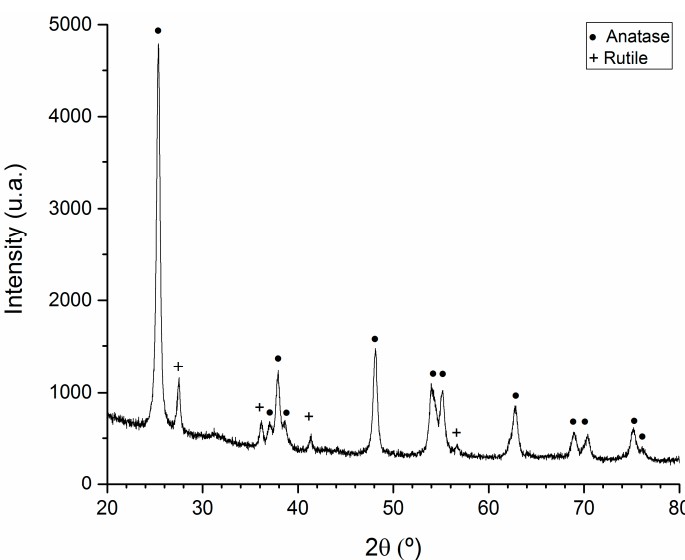

**Figure 2.** XRD diffractogram of the TiO$_2$ NPs used for the fabrication of the sol-gel coatings, which are composed of a mixture of rutile and anatase forms.

Once the nature of TiO$_2$ NPs was corroborated, the next step was a further incorporation into the sol-gel hybrid coating. In Figure 3, it is clearly demonstrated that these metal oxide nanoparticles (TiO$_2$ NPs) have been successfully incorporated into the hybrid sol-gel thin films, because significant differences are observed in the resultant coloration between a coating without TiO$_2$ NPs, which is composed of 6 dips of GPTMS-MTEOS (totally transparent, reference sol-gel coating), and coating with the incorporation of the TiO$_2$ NPs, which is composed of 6 dips of GPTMS-MTEOS-TiO$_2$ NPs (white coloration). In addition, the sol-gel hybrid coatings show a high quality, with a good homogeneity and distribution of the TiO$_2$ NPs across the entire coated surface area.

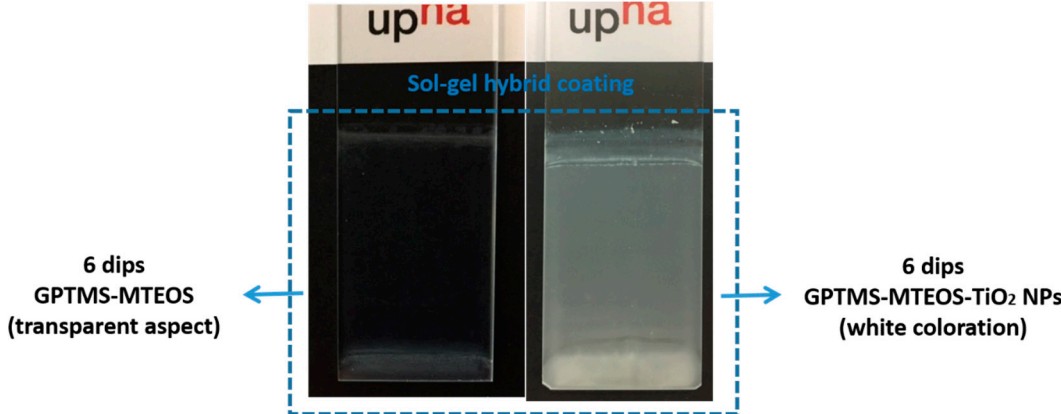

**Figure 3.** Aspect of the hybrid sol-gel coatings without TiO$_2$ NPs (left, totally transparent) and with the presence of TiO$_2$ NPs (right, white coloration) after 6 dips, using a glass slide as a reference substrate.

Once an adequate entrapment of the TiO$_2$ NPs into the sol-gel hybrid coatings was confirmed, the following step was to evaluate the effect of the addition of these metal oxide nanoparticles in the resultant wettability of the coatings for variable thicknesses (1 and 6 dips), as can be observed in Table 2. In this Table, four different samples are analyzed. Sample 1 corresponds to a hybrid coating composed of only GPTMS-MTEOS (1 dip) without the incorporation of any type of additives. Sample 2 corresponds to a hybrid sol-gel matrix composed of GPTMS-MTEOS-TiO$_2$ NPs (1 dip). Samples 3 and 4 correspond to thicker sol-gel hybrid coatings (6 dips) with only GPTMS-MTEOS (sample 3) and GPTMS-MTEOS-TiO$_2$ NPs (sample 4).

**Table 2.** Summary of the different sol-gel hybrid coatings of the study as a function of two experimental variables, the presence of additives (TiO$_2$ NPs) and number of dips (1 and 6), with their corresponding water contact angle (WCA) values.

| Sample | Coating | Number of Dips | Water Contact Angle Value (WCA) |
|--------|---------|----------------|----------------------------------|
| 1 | GPTMS-MTEOS | 1 | $20.66 \pm 1.17$ |
| 2 | GPTMS-MTEOS-TiO$_2$ NPs | 1 | $98.61 \pm 2.11$ |
| 3 | GPTMS-MTEOS | 6 | $20.28 \pm 1.59$ |
| 4 | GPTMS-MTEOS-TiO$_2$ NPs | 6 | $98.51 \pm 2.32$ |

In Figure 4, the variation of the water contact angle (WCA) of the different samples (Figure 4a), as well as their corresponding water droplets images (Figure 4b) can be observed. In addition, Figure 4c shows the aspect of the water droplets on a coated aluminum substrate due to the incorporation of TiO$_2$ NPs in the sol-gel hybrid matrix with a hydrophobic behavior. After observing the experimental values of the water contact angle (WCA), two main conclusions can be obtained. The first is that the presence of TiO$_2$ NPs allows a significant increase in the resultant wettability of the sol-gel hybrid coatings. For example, the WCA value is increased from $20.66 \pm 1.17°$ (sample 1) up to $98.61 \pm 2.11°$ (sample 2) for a thickness coating of 1 dip. Furthermore, an increase in the number of dips from 1 (sample 2) to 6 dips (sample 4) in the samples with TiO$_2$ NPs has not obtained an improvement in the resultant hydrophobicity of the sol-gel hybrid coatings, showing for both cases similar WCA values ($98.61 \pm 2.11°$ and $98.51 \pm 2.32°$, respectively).

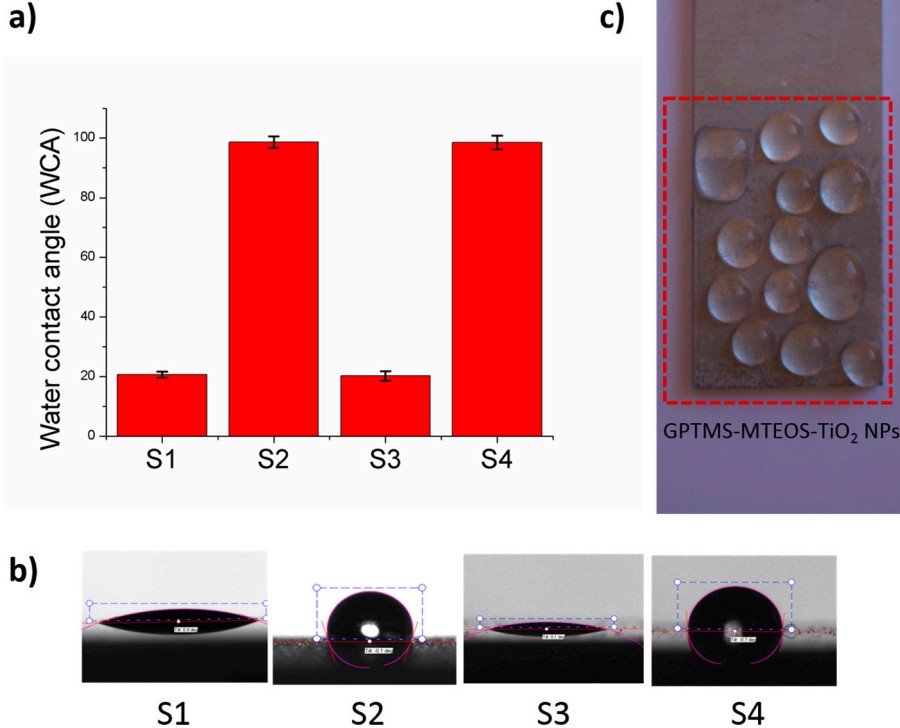

**Figure 4.** Variation of the water contact angle values (**a**) and aspect of the water droplets (**b**) for S1, S2, S3, and S4 samples. Water droplet images on hybrid sol-gel coating based on TiO$_2$ NPs (**c**).

In order to enhance the hydrophobic behavior of the sol-gel hybrid matrices, a new second coating (6 dips) based on the incorporation of a new fluorinated metal alkoxide precursor (PFAS) was performed on the previous fabricated sol-gel hybrid (GPTMS-MTEOS-TiO$_2$ NPs) matrix. This combination of uniformly distributed TiO$_2$ NPs with a new metal-alkoxide precursor with fluorinated polymeric chains (PFAS) in the outer surface, allows an improvement of the hydrophobic property of the sol-gel hybrid coatings [30,42], as can be observed in the corresponding WCA values presented in Table 3.

**Table 3.** Summary of the different sol-gel hybrid coatings with their corresponding water contact angle (WCA) values before and after thermal treatment with the incorporation of PFAS metal alkoxide precursor.

| Coating | Number of Dips | Thermal Treatment | Water Contact Angle Value (WCA) |
|---|---|---|---|
| GPTMS-MTEOS | 6 | No | 20.28 ± 1.59 |
| GPTMS-MTEOS-TiO$_2$ NPs | 6 | No | 98.51 ± 2.32 |
| (GPTMS-MTEOS-TiO$_2$ NPs) + PFAS | 6 | No | 107.11 ± 1.75 |
| (GPTMS-MTEOS-TiO$_2$ NPs) + PFAS | 6 | Yes | 120.76 ± 2.17 |

Another significant result was that the thermal treatment at 180 °C for 4 h (curing step), performed in the samples with PFAS precursor, produced an increase of the resultant WCA value from 107.11 ± 1.75° (room condition) up to 120.76 ± 2.17° (after thermal treatment). This curing step was used to form a dense three-dimensional film structure by thermally induced self-condensation reactions within the coating material that remove hydroxide groups from the remaining silanol molecules [43,60,61]. The WCA value was increased by heat treatment, because a decrease of the hydrophilic hydroxyl end groups (-OH) was obtained due to a high degree of cross-linking reactions. In addition, it is worth mentioning that no higher temperatures than 300 °C were used, because this could result in a chemical decomposition of the perfluoroalkyl groups of the PFAS with a loss of

the hydrophobic behavior of the sol-gel coatings [45]. The incorporation of PFAS into the previous hybrid matrix composed of TiO$_2$ NPs and a further thermal treatment produced an increase in the wettability from $98.51 \pm 2.32°$ up to $120.76 \pm 2.17°$, which corresponds to an improvement of 22.5% in the hydrophobic property of the sol-gel coatings. In Figure 5, this continuous increase in hydrophobic property (WCA images with their corresponding results) as a function of the presence of PFAS and the thermal treatment for inducing the chemical cross-linking can be observed.

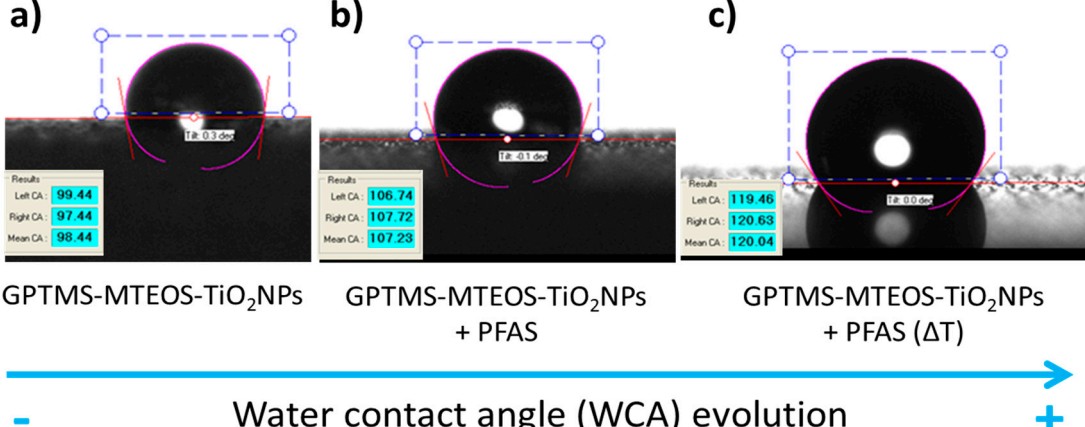

**Figure 5.** Comparison between the water contact angle images recorded for the films obtained when aluminum substrates are coated with GPTMS-MTEOS-TiO$_2$NPs (**a**), GPTMS-MTEOS-TiO$_2$NPs-PFAS (**b**), and GPTMS-MTEOS-TiO$_2$NPs-PFAS after thermal treatment (**c**).

After the evaluation of the simultaneous and positive effect of three different factors, namely the incorporation of TiO$_2$ NPs, the use of the PFAS precursor in the outer surface, and a thermal treatment (curing step at 180 °C), the wettability of the sol-gel hybrid films with an important enhancement in the hydrophobic property was observed. In addition, two mechanical tests, namely the adhesion of the coatings as well as pencil hardness, were undertaken in order to appreciate any important difference in the resultant mechanical properties of the coatings between GPTMS-MTEOS (sample A), GPTMS-MTEOS-TiO$_2$NPs (sample B), and thermally treated GPTMS-MTEOS-TiO$_2$NPs-PFAS (sample C), and the results can be observed in Figure 6. According to the adhesion tests (ASTM D3359), the experimental results indicated that both sol-gel hybrid coatings showed a very good adhesion with the aluminum substrate. It was observed that the edges of the cuts are completely smooth with none of the squares of the lattice detached. All the tests have shown 0% removal of the coatings from the substrate, which corresponds to the best rating of 5B. However, after performing pencil hardness tests (ASTM D3363), it was found that the pencil hardness values were higher for the samples with TiO$_2$NPs in comparison with the reference GPTMS-MTEOS. The pencil hardness of GPTMS-MTEOS without metal oxide nanoparticles (sample A) was 1H, whereas the presence of TiO$_2$ NPs in this hybrid matrix (sample B) resulted in a positive effect, because an enhancement in the pencil hardness value was observed up to a value of 3H. In addition, the combination of fluorinated precursor with a curing heat treatment (sample C) shows a greater effect, with a pencil hardness value of 5H. This curing step increases the degree of cross-linking between the alkoxides, rendering the film denser and harder, and as result, a significant enhancement in the durability of the coating films can be obtained [43,45,60].

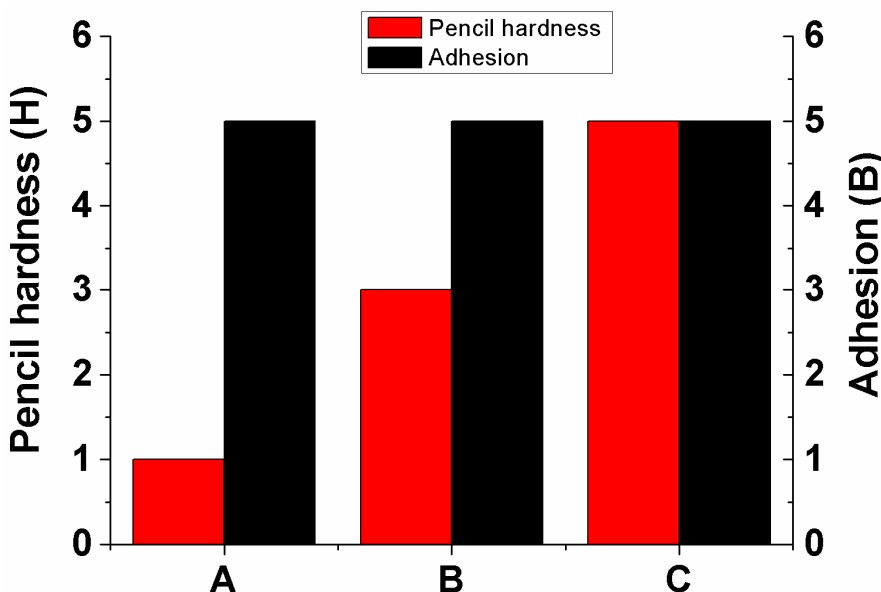

**Figure 6.** Experimental results for pencil hardness (red column) and adhesion tests (black column) for sample A (GPTMS-MTEOS), sample B (GPTMS-MTEOS-TiO$_2$NPs), and sample C (GPTMS-MTEOS-TiO$_2$NPs-PFAS) after thermal treatment.

Another important aspect related to the presence of both TiO$_2$ NPs and PFAS in the sol-gel hybrid coatings involves the analysis of their corresponding impact on the resultant local corrosion susceptibility (pitting corrosion potential), with the aim of designing novel coatings with good corrosion protective properties. As can be observed in Table 4, the pitting corrosion potential for different sol-gel samples is analyzed in order to establish a direct relationship between both hydrophobic behavior and corrosion resistance susceptibility when the samples are immersed in a 3.5% NaCl solution. First of all, it should be noted that the best result is observed for the sample composed of GPTMS-MTEOS-TiO$_2$NPs-PFAS after thermal treatment, showing the most positive pitting potential in comparison with the other samples. In addition, in all the cases of this study, there is a shift of the pitting corrosion potential to a more positive potential in comparison with the aluminum bare substrate (−0.57 V). This effect is firstly observed for the samples with the hybrid matrix without metal oxide nanoparticles, such as GPTMS-MTEOS and GPTMS-MTEOS-PFAS. This result clearly indicates that a hydrophobic surface provided with a PFAS precursor can be employed to improve the corrosion resistance of the coatings. Furthermore, the combination of both TiO$_2$ NPs and the PFAS precursor with thermal treatment during the curing step has shown the best results, because the pitting corrosion potential is shifted to the most positive potential value from −0.45 V (room temperature) to −0.40 V (after thermal treatment).

**Table 4.** Summary of the pitting corrosion potential values obtained as a function of two experimental variables: number of dips and thermal treatment (curing step).

| Coating | Number of Dips | Thermal Treatment | Pitting Corrosion Potential |
|---|---|---|---|
| Aluminum bare substrate | - | - | −570 mV |
| GPTMS-MTEOS | 6 | No | −520 mV |
| (GPTMS-MTEOS) + PFAS | 6 | No | −480 mV |
| (GPTMS-MTEOS-TiO$_2$ NPs) + PFAS | 6 | No | −450 mV |
| (GPTMS-MTEOS-TiO$_2$ NPs) + PFAS | 6 | Yes | −400 mV |

The main conclusion that can be derived from these experimental results is that the hybrid coating composed of TiO$_2$ NPs shows a better blocking effect of chloride ions which flow from the solution

to the metal surface. This result indicates that a multilayer structure system based on the gradual and successive incorporation of $TiO_2$ NPs combined with the PFAS precursor (6 dips) improves the corrosion resistance of the aluminum substrate. In Figure 7, one can see the cyclic polarization curves plotted for the aluminum bare substrate, GPTMS-MTEOS, GPTMS-MTEOS-$TiO_2$NPs-PFAS, and GPTMS-MTEOS-$TiO_2$NPs-PFAS after thermal treatment. In addition, another interesting result is that the number of corrosion pits in the aluminum bare substrate is higher in comparison with the sol-gel hybrid coatings. Finally, these results combined with the previous observed results in both Figures 5 and 6 can be used to confirm that the presence of these $TiO_2$ NPs in the hybrid sol-gel matrices allow the design of corrosion resistance coatings with better mechanical properties, as well as a higher hydrophobic behavior in comparison with blank sol-gel coatings.

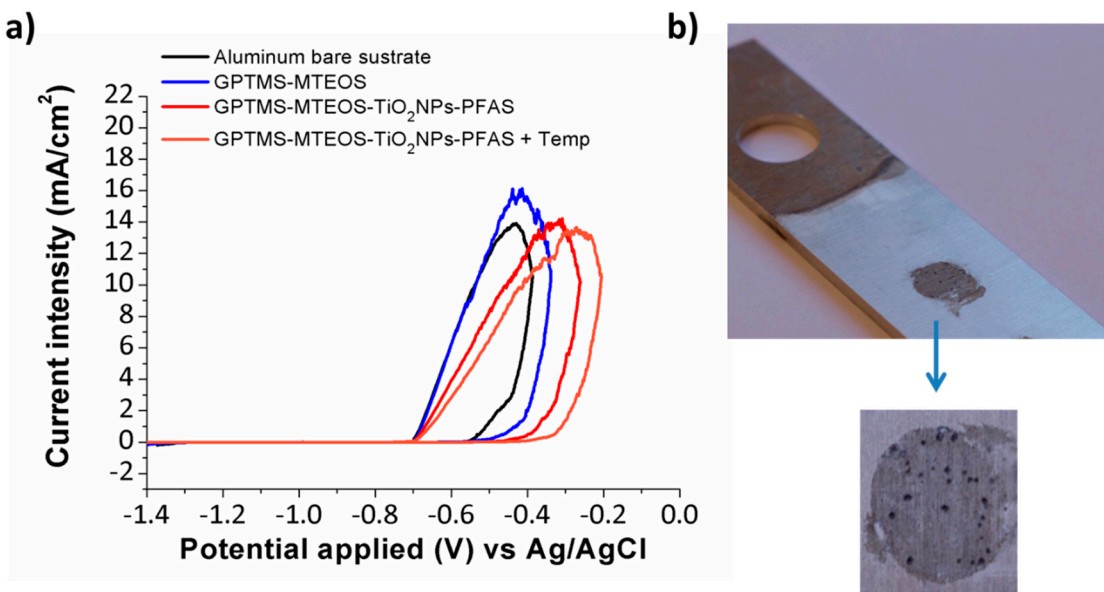

**Figure 7.** (**a**) Cyclic polarization curves in 3.5% NaCl solution for aluminum bare substrate (black line), 6 dips of GPTMS-MTEOS coating (blue line), 6 dips of GPTMS-MTEOS-$TiO_2$NPs-PFAS coating (red line), and 6 dips of GPTMS-MTEOS-$TiO_2$ NPs-PFAS coating after thermal treatment (orange line). (**b**) Aspect of the coating after performing the sol-gel hybrid coatings, and a zoom of the surface area with corresponding pitting marks.

In order to demonstrate an adequate incorporation of $TiO_2$ NPs along the hybrid sol-gel matrix, as well as the presence of fluorine groups in the outer surface which are able to show a water-repellent behavior in the outer surface, glow discharge optical emission spectrometry (GD-OES) analysis was performed on the sample composed of 6 dips of GPTMS-MTEOS-$TiO_2$NPs-PFAS after thermal treatment, as it showed the best results as both a corrosion and mechanical resistance coating. This analysis allows a determination of the profile concentration related to all the chemical elements which are present in the hybrid sol-gel coating, as can be observed in Figure 8.

In this figure, the limit between the sol-gel hybrid coating and the beginning of the bare aluminum substrate can be clearly differentiated. The titanium profile clearly indicates that a successive and gradual incorporation of $TiO_2$ NPs has been successfully embedded in the sol-gel coating during the deposition process, because titanium is observed on the outer surface as well as the inner part of the coating. Another interesting result observed in this figure is that the outer part of the sol-gel hybrid coating presents a smaller concentration of carbon (red line), whereas the fluorine concentration is higher in the outer surface. This indicates that the fluorinated chains are disposed to the exterior part of the coating in order to impart hydrophobicity. In addition, the presence of other specific chemical elements is also observed, such as carbon, oxygen and silica related to the chemical structure of the metal alkoxides (GPTMS, MTEOS, and PFAS) used for the fabrication of the hybrid sol-gel

matrix. Other important information obtained by the GD-OES analysis is that the total thickness is approximately 2 μm when the aluminum profile is close to 100% (bare substrate).

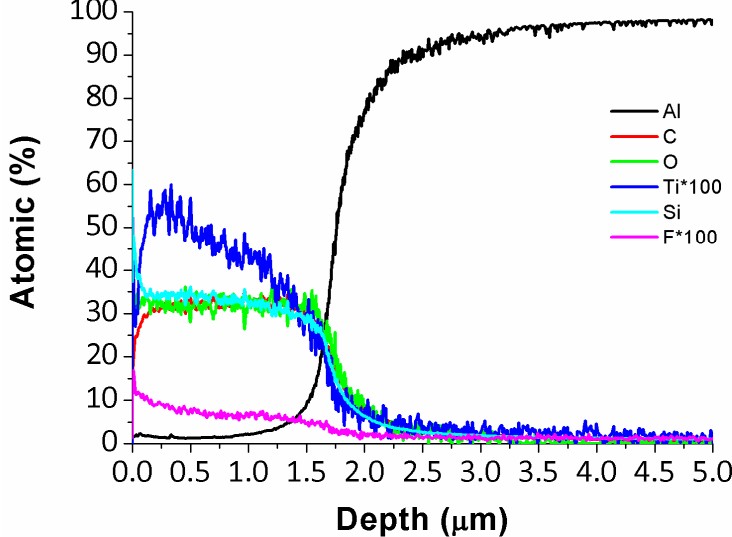

**Figure 8.** A full view of glow discharge optical emission spectrometry (GD-OES) concentration profiles for a depth of 5 μm in the sample composed of GPTMS-MTEOS-TiO$_2$NPs-PFAS ($\Delta T$) with the presence of aluminum (black line), carbon (red line), oxygen (green line), titanium (blue line), silicon (cyan line), and fluorine (pink line). Note that the titanium and fluorine concentration has been multiplied by a factor of 100 to be visible on the graph.

In order to have a better appreciation of the resultant thickness, as well as the resultant surface morphology of the sol-gel hybrid coating, a FE-SEM analysis has been performed on both the fractured cross-section and the outer surface of the sample, as can be seen in Figure 9. First of all, it can be clearly observed that the outer perfluoroalkyl group modified sol-gel hybrid coating shows a microporous morphology (Figure 9a) throughout the surface, resulting in an enhancement of surface roughness. In addition, the Energy-dispersive X-ray (EDX) analysis of the coating surface confirms the presence of chemical elements (F, O, or Si) derived from the PFAS precursor. After observing the cross-section SEM image (Figure 9b), it was found that the average thickness of the sol-gel hybrid coating was approximately 1.75–2 μm, which is in concordance with the GD-OES analysis obtained in Figure 8.

In order to have a better characterization of the resultant morphology of the sol-gel hybrid coatings, SEM and AFM analyses were performed. As previously commented, in Figure 10 the SEM images of the topographic surface are presented, which clearly indicate a microporous morphology throughout the surface of the aluminum substrate, because the presence of variable microgrooves can be appreciated. These microgrooves have the tendency to entrap air in the pores of the films, which contributes to the beading up and easy rolling of the water droplets over the protrusions, making an enhancement in the water repellency possible, with a higher water contact angle value [44,61,62]. The use of PFAS in the outer surface makes an increase in the surface roughness possible (corroborated by SEM images), which is one of the most important requirements to achieve a hydrophobic character, along with the low interfacial energy. According to this, the surface energy of the functional groups is decreased in the following order –CF$_3$ < –CF$_2$H < –CF$_2$ < –CH$_3$ < –CH$_2$, hence the use of the PFAS precursor in the outer surface, which is composed of fluorinated functional groups (CF$_3$ and CF$_2$), enables a considerable enhancement in the hydrophobic and water repellent properties [63,64].

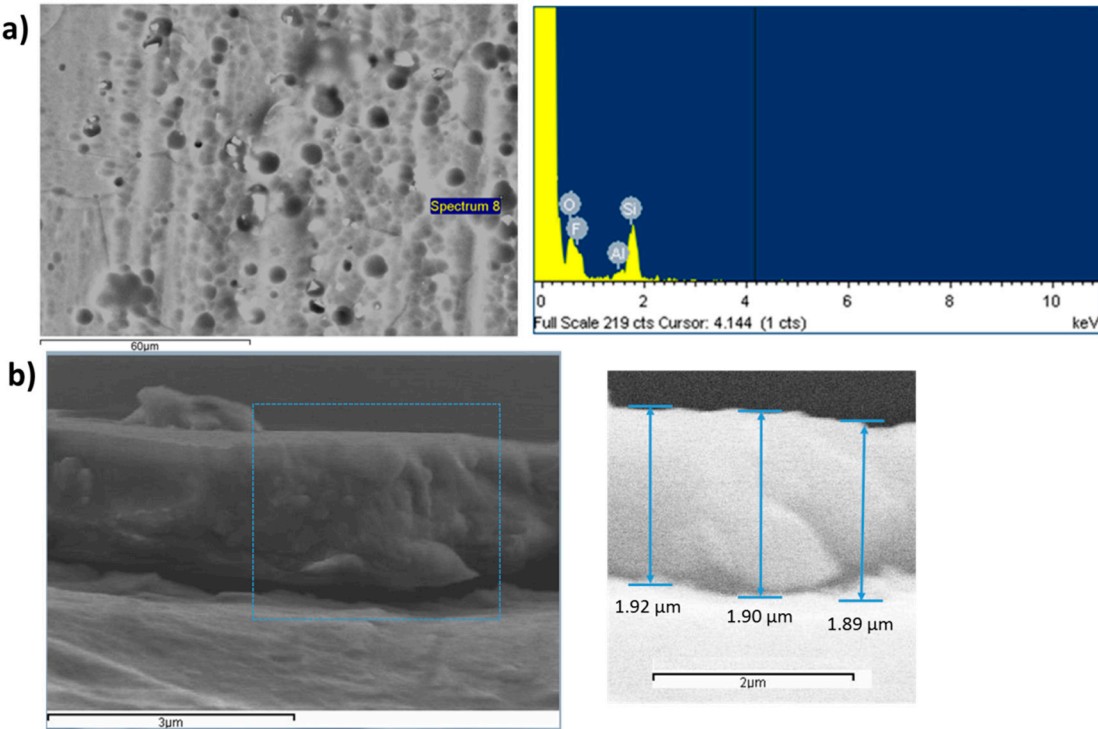

**Figure 9.** SEM images of the outer surface with its corresponding Energy-dispersive X-ray EDX analysis (**a**) and the cross-section of the sol-gel hybrid coating on the aluminum substrate (**b**).

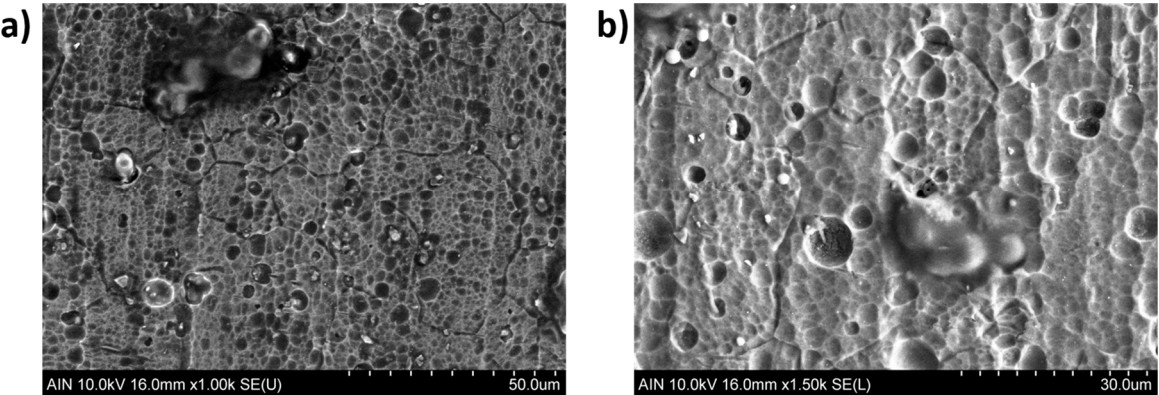

**Figure 10.** SEM images of the topographic surface of the sol-gel hybrid coatings for a scale bar of 50 μm (**a**) and 30 μm (**b**).

In addition, AFM analysis was performed for the characterization of the topographic surface of the sol-gel hybrid coating. In Figure 11, a two dimensional (2D) height image (Figure 11a), three dimensional (3D) topographic surface (Figure 11b), and the section profile in three different regions of the surface (Figure 11c) for a scan area of 20 μm × 20 μm, are presented. From the AFM images of Figure 11a,b, a microscale roughness can be observed, whereas from the profile section (Figure 11c), a variable number of protrusions (hills) can be seen along the topographic surface of three different regions of analysis. This morphological evidence associated with the terminal perfluoro groups in the outer surface of the sol-gel coatings allows an entrapment of enough air to prevent the penetration of water into the protrusions, making a reduction in the contact area of the water droplets with the surface possible, and as a result, an improvement in the water repellency behavior is obtained [43,44,65]. To sum up, the use of this PFAS precursor for the fabrication of the hybrid sol-gel coatings in the outer part of the coating makes a dual synergetic effect possible; that is, a considerable enhancement in the surface roughness of the coating as well as a lowering of the surface energy.

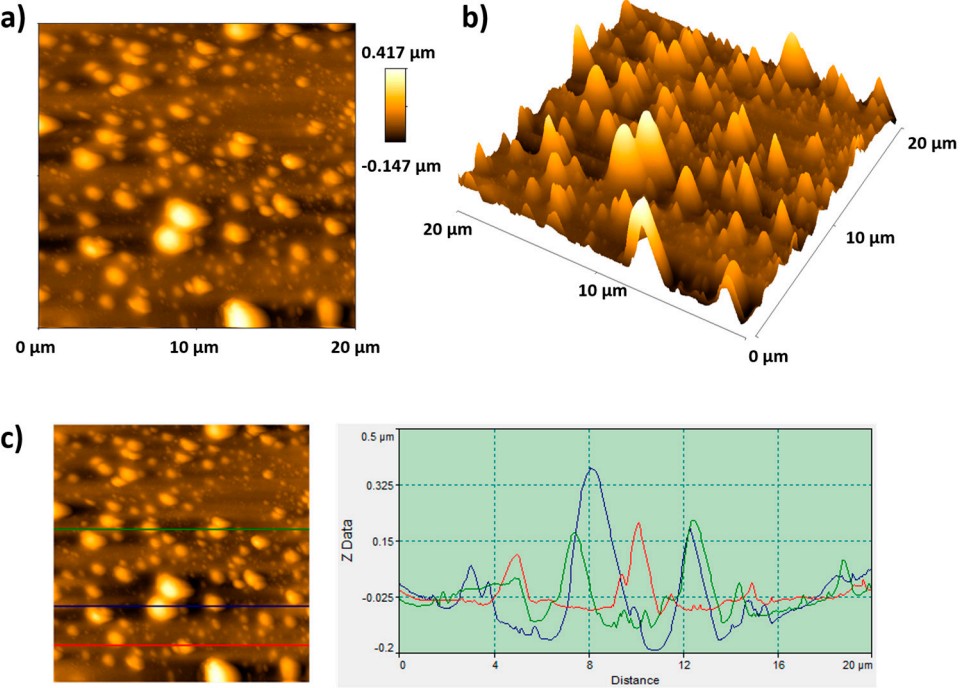

**Figure 11.** Atomic force microscopy (AFM) images in tapping mode of the resultant topographic surface of the sol-gel hybrid coatings in 2D (**a**), 3D (**b**), and the resultant section profiles in three different locations of the coatings (**c**).

Finally, the corrosion behavior of the sol-gel hybrid coatings was also analyzed using electrochemical impedance spectroscopy (EIS) in 3.5% NaCl aqueous solution, and the data processed by means of the Z-View software (Metrohm, Herisau, Switzerland). The experimental results show that there is a direct relationship between both the hydrophobic property and corrosion resistance, as can be appreciated in the Bode plots when the sol-gel coated substrates have been exposed for a specific period of 168 h (see Figure 12). In this sense, it was observed that the impedance is increased by an order of magnitude for the sample composed of 6 dips of GPTMS-MTEOS-TiO$_2$NPs-PFAS (the highest WCA value) in comparison with the sol-gel coated samples without PFAS precursor, such as GPTMS-MTEOS. These results clearly suggest that the presence of a hydrophobic layer in the outer surface acts as a physical diffusion barrier due to the entrapment of air between the roughness protrusions [40,41,45]. In addition, another interesting aspect is that the samples loaded with TiO$_2$NPs (GPTMS-MTEOS-TiO$_2$NPs) show a higher impedance value in comparison with the samples without TiO$_2$NPs (sol-gel blank solution composed of only GPTMS-MTEOS). The presence of these TiO$_2$ NPs has a beneficial impact on the sol-gel hybrid coating resistance, since the impedance values at low frequency are increased, and as a result, an enhancement of the barrier properties is obtained if it is compared with the sol-gel blank coating. In addition, it is also demonstrated that a thicker coating (6 dips) has a better corrosion resistance than a thinner coating (1 dip), because the resultant impedance value is higher for the samples coated with a higher number of dips (thicker coatings).

To sum up, the experimental results from EIS measurements clearly demonstrate that the combination of both metal oxide nanoparticles (TiO$_2$ NPs) and the use of PFAS precursor in the outer surface for increasing the hydrophobicity allows the design of novel coatings, which can considerably reduce the corrosion susceptibility, while showing very good mechanical properties due to the presence of these metal oxide nanoparticles inside the sol-gel hybrid coatings.

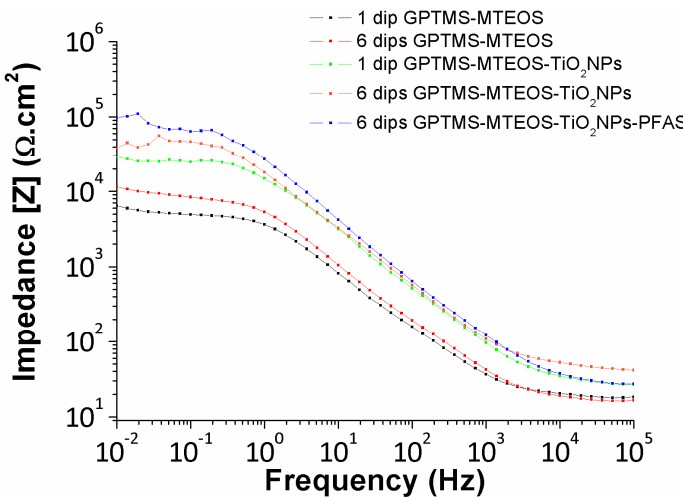

**Figure 12.** Bode plots of several sol-gel hybrid coatings (GPTMS-MTEOS, GPTMS-MTEOS-TiO$_2$NPs, and GPMTS-MTEOS-TiO$_2$NPs-PFAS) as a function of the number of dips after immersion in 3.5% NaCl solution for a fixed period of 168 h.

## 4. Conclusions

In this work, the design of a novel sol-gel hybrid coating with the dual properties of corrosive protection and mechanical durability is presented. A hybrid matrix has been fabricated by the copolymerization of two organic modified silica alkoxides, namely 3-(glycidyloxypropyl)trimethoxysilane (GPTMS) and methyltriethoxysilane (MTEOS). This hybrid matrix plays a key role because it is used for a further entrapment of metal oxide nanoparticles (TiO$_2$ NPs). It has been demonstrated that the presence of these TiO$_2$ NPs produces a surface modification with an enhancement of the wettability of the coating, showing higher water contact angle (WCA) values. However, in order to increase the resultant hydrophobicity of the coatings, a new silica precursor based on fluorinated polymeric chains (PFAS) has been added to the previous sol-gel hybrid matrix. The results reveal that the sol-gel coatings have shown an effective modification, because an important improvement of the hydrophobicity has been observed. In addition, the pencil hardness test has shown a considerable enhancement of the mechanical characteristics due to the thermal post-treatment (curing step). In order to evaluate the corrosion resistance of the sol-gel hybrid coatings, potentiodynamic polarization tests, as well as electrochemical impedance spectroscopy have been performed. Finally, the experimental results clearly demonstrate that the combination of both metal oxide nanoparticles (TiO$_2$ NPs) in the inner part of the coating and PFAS precursor in the outer surface enables the development of novel coatings with good mechanical durability, which can reduce the corrosion susceptibility by showing a good protective barrier for longer periods of time.

**Author Contributions:** Conceptualization, P.J.R. and J.D.M.; Methodology, P.J.R. and J.D.M.; Software, P.J.R. and C.B.; Validation, P.J.R., A.M., and R.R.; Formal Analysis, P.J.R., J.F.P. and R.R.; Investigation, P.J.R. and J.D.M.; Resources, P.J.R., A.M. and J.F.P.; Data Curation, P.J.R. and J.D.M.; Writing-Original Draft Preparation, P.J.R. and J.D.M.; Writing-Review & Editing, P.J.R., J.F.P. and R.R.; Visualization, P.J.R. and C.B.; Supervision, P.J.R. and R.R.

**Funding:** This research was funded by Public University of Navarre and the APC was funded by Project PRO-UPNA18–6107–FRRRIO.

**Acknowledgments:** This work was supported by the Spanish Economy and Competitiveness Ministry, FEDER (Project TRA2013-48603-C4-1-R-HELADA), and by the Public University of Navarre (Project PRO-UPNA18–6107–FRRRIO). The authors would like to express their gratitude to Nadetech Inc. for the tune-up of the robot used for the deposition of sol-gel hybrid coatings.

**Conflicts of Interest:** The authors declare no conflict of interest.

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
