# Peer review of "Hydrophobic and Corrosion Behavior of Sol-Gel Hybrid Coatings Based on the Combination of TiO2 NPs and Fluorinated Chains for Aluminum Alloys Protection"

_metals, doi:10.3390/met8121076_

Round 1
Reviewer 1 Report
The paper by Rivero and al. reports on the coating of a kind of an aluminum-based alloy by using the dip-coating technique and a step-by-step approach. Although the nature of the coatings is uncertain (TiO2 nanoparticles with uncertain structure and composition), improved anticorrosive and hydrophobic properties have been verified by water contact angle, impedance spectroscopy, optical microscopy measurements. Furthermore, the language does not reach the qualitystandard. By considering that the MS leaks of a blend of issues, including the originality (plagiarism) and a few fundamental aspects, cannot be considered for publication in the current state. The authors are urged to critically revise in detail the text and scientific issues, addressing the following points (listed here in a priority viewpoint).
1). Some sentences, in particular: i) in the 1 introduction, ii) 2.4 – electrochemical corrosion test sections, appear the same in previous papers (see referenced in source 1 and 2, in the attached plagiarism report).
2). The nature of the TiO2 layer is uncertain and a more detailed clarification is needed. The only reference to the structure is in in the introduction: “These nanoparticles are prepared in anatase or rutile forms and 79 they can be used for increasing the resultant surface roughness of the coatings.” (line79, Page2). No more clarification is given in the discussion of the prepared samples. That is the nature of the TiO2 nanoparticles, anatase, rutile, mixed composition or what? Please, provide a dedicated characterization (i.e. XRD or Raman).
3). I also think that for improving the quality of the paper, the correlation of the observed wettability properties should be better addressed, for example, with the top morphology (acquired by higher resolution SEM images of the Top layer / AFM images or surface chemistry properties (i.e. IR taken on the surface).
4). With particular reference to coatings with TiO2 nanoparticle and the related wettability property, the authors give their main focus to protection corrosion resistance issues, but there are also some other applications where such type of coating have been addressed, including photocatalysis, bacteriostatic activity, etc. I think relevant to include a general comment about these kinds of properties. The authors can refer to: Cravanzola, S., RSC Advances 5, 2015, 103255; Liu J., RSC Advances 7, 2017, 43938; De Falco G., Surface and Coatings Technology 349, 2018, 830
(2015). "Development of a multifunctional TiO2/MWCNT hybrid composite grafted on a stainless steel grating.".
5). Grammar/English:
“Other interesting aspect related to the sol-gel technology …” (line 61, page 2). Please, correct.
“In addition, the simplicity, versatility and easiness of the fabrication process by using the dip-coating technique makes possible the development of effective …” (line 97, page 3). Please, correct the verb.
“This combination of uniform distributed TiO2 …” (line 220, page 6). Please substitute “uniform” with an adverb.
“As it can be observed in Table 3, the pitting corrosion potential for different sol-gel samples are analysed …” (line 276, page 8). Please, correct the verb.
“In addition, the presence of other specific chemical elements are also observed …” (line 331, page 11). Please, correct the verb.
“Finally, the corrosion behaviorof the sol-gel hybrid coatings have been also analyzed” (line 351, page 12). Please, correct the verb.
“It has been demonstrated that the presence of these TiO2NPsproduce a surface modification” (line 384, page13). Please, correct the verb.
“In addition, the importance of a further thermal treatment
(curing step) is also studied because a considerable enhancement of the
mechanical properties have been obtained”
(line 389, Page 13). Please rewrite the sentence and correct the verb.

Author Response
Comments from Reviewer_1:
The paper by Rivero and al. reports on the coating of a kind of an aluminum-based alloy by using the dip-coating technique and a step-by-step approach. Although the nature of the coatings is uncertain (TiO2 nanoparticles with uncertain structure and composition), improved anticorrosive and hydrophobic properties have been verified by water contact angle, impedance spectroscopy, optical microscopy measurements. Furthermore, the language does not reach the quality standard. By considering that the MS leaks of a blend of issues, including the originality (plagiarism) and a few fundamental aspects, cannot be considered for publication in the current state. The authors are urged to critically revise in detail the text and scientific issues, addressing the following points (listed here in a priority viewpoint).
First of all, we would like to thank to the anonymous reviewer for his/her positive criticisms. All the changes have been highlighted in yellow for a better understanding and localization of them, as it can be appreciated in the revised version of the manuscript. In addition, this revised version has been rewritten in order to clarify important key aspects for the reader.
The Introduction section has been modified and new paragraphs have been added with the aim to show three critical aspects which can make this work more attractive. The first one has been focused on the importance of aluminum coatings in some advanced technologies for energy production and storage (batteries or capacitors). The second one has been focused on the great multifunctionality related to the use of TiO2 NPs and their importance in other fields such as photocatalysis, photovoltaic, antibacterial activity or self-cleaning applications. And the third one has been focused on showing the advantages of using the sol-gel technology for corrosion protection, making a comparative table of the different alkoxide precursors used as well as the presence of specific corrosion inhibitors.
A more exhaustive topographical characterization by using AFM and SEM analysis has been performed in order to explain the correlation between the observed wettability properties and the resultant surface morphology. Moreover, XRD analysis has been also performed in order to clarify the nature of the TiO2 NPs used for the fabrication of the sol-gel hybrid coatings. Other aspect to remark is that the bibliography section has been also updated with a more recent and actual list of references, as it can be appreciated in this new version of the manuscript.
Finally, a native English speaking colleague with skills in thin-films fabrication has thoroughly revised the manuscript in order to improve the quality standard of this paper. We hope that this new revised version of the manuscript can be published in the Journal of Metals.
1) Some sentences, in particular: i) in the 1 introduction, ii) 2.4 – electrochemical corrosion test sections, appear the same in previous papers (see referenced in source 1 and 2, in the attached plagiarism report).
According to the originality of the paper and after checking the corresponding plagiarism report, we can appreciate that two references (with 12% and 6% of similarity, respectively) are from our own research group (Materials Engineering Lab, Public University of Navarra). It is important to remark that the first work (12% similarity) is devoted to the fabrication of anticorrosive surfaces and due to this, the experimental corrosion tests are the same and the description of the electrochemical corrosion tests in the experimental section are identical. Due to this, the experimental section related to the electrochemical corrosion tests has been completely modified in order to avoid the use of the same sentences.
According to the second work (6% similarity) has not been cited in the manuscript in order to avoid the auto-citation because the fabrication of the coating is based on the electrospinning process instead of sol-gel technology. However, this work has been cited in the revised version for using the same fluorinated precursor for obtaining a hydrophobic surface, although it is deposited by Chemical Vapor Deposition (CVD). In this work, we have observed similar sentences in the Introduction section related to the use of aluminum alloys as well as their advantages. Due to this, the Introduction section has been rewritten and new references have also incorporated related to the use of aluminum in the development of new technologies for energy production and storage (batteries or capacitors).
Changes on the “Introduction” section
“Aluminum and its alloys are, after steel and cast irons, the group of metallic materials most widely used by the industry. Their unique combination of low density, mechanical properties, high thermal and electrical conductivity and affordable cost make aluminum the choice material for an increasing number of applications such as airframes, automotive components, power lines, heat exchangers or food containers”.
“Despite of the low strength and high ductility of the pure metal, many of its alloys such as those of the 2XXX, 6XXX and 7XXX series can reach excellent mechanical properties through solution and ageing heat treatments [1]. The AA6061 is a paradigmatic example, being one of the most employed alloys in aeronautics as well as in the automobile industry or in shipbuilding. In addition to the above mentioned applications, aluminum is also playing a major role in the development of new technologies for energy production and storage. Bipolar plates employed in PEM fuel cells are perhaps the best studied example of aluminum based components working in a chemically aggressive environment [2, 3]. On the other hand, among energy storage systems where aluminum electrodes are a key element it is worth mentioning capacitors [4] and new batteries [5-7]”.
“For structural applications, in environments where chloride ions can be present, the use of chromates as corrosion inhibitors has been the standard solution for providing the required protection in aqueous media. In the case of aluminum alloys this can be implemented chromic acid anodized and chromate conversion coatings [10, 11]. However, in the last two decades, the use of hexavalent chromium compounds has been banned in most industrialized countries because they are extremely dangerous for the ecosystem and human health [12, 13]. Due to these serious drawbacks, the scientific community has been looking for safer and ecofriendly alternatives with the aim of replacing these toxic chromium compounds [14]. In the case of AA6061 bipolar plates, PANI and PPy films [2] as well as PVD coatings for the AA5083 [15] have been suggested as a possible solution for preventing corrosion in PEM fuel cells”.
“Scalability must be always a main concern in the choice of solutions for these new problems. Sol-gel coating technology is susceptible of being scaled by implementing spray coating methods or, in particular cases, roll-to-roll processes. In addition, sol-gel technology is a potential candidate for corrosion protection because it shows interesting features such as a good barrier effect, possibility of incorporating inhibitors into the matrix, good adhesion over different metallic substrates and good compatibility with additional top layers”.
Changes of the “Electrochemical corrosion tests” section
“All the corrosion tests were conducted through cyclic potentiodynamic polarization techniques and electrochemical impedance spectroscopy (EIS) by using an Autolab PGSTAT30 galvanostat/potentiostat system at room temperature. The experiments were conducted with three electrode system composed of sol-gel coated aluminum substrate as working electrode (WE), platinum wire as a counter electrode (CE) and an Ag/AgCl electrode as the reference electrode (RE). It should be noted that before testing, all the samples of study were exposed for a while (1 hour) with the intention of stabilizing the open circuit potential (OCP)”.
“Cyclic potentiodynamic polarization measurements were performed in order to determine the localized corrosion susceptibility. Samples were immersed in the saline solution (3.5 wt.% in ultrapure water) for a period of time of 1 hour before initiating polarization and after that, the potential scan started from Ecorr. The scan rate was of 0.6V/h towards the noble direction. Once the current density reached a specific value of 5 mA/cm2, the scan direction was reversed until the hysteresis loop was closed or until the corrosion potential was reached”.
“Electrochemical impedance spectroscopy measurements were performed in 3.5 wt.% NaCl solution in a frequency range of 0.01 Hz to 100 kHz with a wave amplitude of 10 mV at room temperature”.
Changes on the “Bibliography” section
The following references have been added in the revised manuscript:
(2) Antunes, R. A.; Oliveira, M. C. L.; Ett, G.; Ett, V. Corrosion of Metal Bipolar Plates for PEM Fuel Cells: A Review. Int J Hydrogen Energy 2010, 35, 3632-3647.
(3) Deyab, M. A. Corrosion Protection of Aluminum Bipolar Plates with Polyaniline Coating Containing Carbon Nanotubes in Acidic Medium Inside the Polymer Electrolyte Membrane Fuel Cell. J. Power Sources 2014, 268, 50-55.
(4) Fan, Z.; Zhu, C.; Chang, F. Effect of Thiourea on the Corrosion Behavior of Aluminum Anode Foil of Aluminum Electrolytic Capacitor. J Mater Eng Perform 2018, 27, 4168-4175.
(5) Watanabe, M.; Dokko, K.; Ueno, K.; Thomas, M. L. From Ionic Liquids to Solvate Ionic Liquids: Challenges and Opportunities for Next Generation Battery Electrolytes. Bull. Chem. Soc. Jpn. 2018, 91, 1660-1682.
(6) Yamada, Y.; Chiang, C. H.; Sodeyama, K.; Wang, J.; Tateyama, Y.; Yamada, A. Corrosion Prevention Mechanism of Aluminum Metal in Superconcentrated Electrolytes. ChemElectroChem 2015, 2, 1687-1694.
(7) Wang, R.; Li, W.; Liu, L.; Qian, Y.; Liu, F.; Chen, M.; Guo, Y.; Liu, L. Carbon black/graphene-Modified Aluminum Foil Cathode Current Collectors for Lithium Ion Batteries with Enhanced Electrochemical Performances. J Electroanal Chem 2019, 833, 63-69.
(15) Barranco, J.; Barreras, F.; Lozano, A.; Lopez, A. M.; Roda, V.; Martin, J.; Maza, M.; Fuentes, G. G.; Almandoz, E. Cr and Zr/Cr Nitride CAE-PVD Coated Aluminum Bipolar Plates for Polymer Electrolyte Membrane Fuel Cells. Int J Hydrogen Energy 2010, 35, 11489-11498.
2) The nature of the TiO2 layer is uncertain and a more detailed clarification is needed. The only reference to the structure is in the introduction: “These nanoparticles are prepared in anatase or rutile forms and they can be used for increasing the resultant surface roughness of the coatings.” (line79, Page2). No more clarification is given in the discussion of the prepared samples. That is the nature of the TiO2 nanoparticles, anatase, rutile, mixed composition or what? Please, provide a dedicated characterization (i.e. XRD or Raman).
According to the demand from the reviewer, XRD analysis has been performed in order to clarify the nature of the TiO2 NPs. As it can be appreciated in the new Figure 1, the XRD diffractogram indicates that the resultant titanium oxide used for the fabrication of the sol-gel hybrid coatings consists of a mixture of rutile and anatase because both phases can be clearly appreciated.
A new subsection in the Experimental section has been incorporated in the revised version and the XRD diffractogram as well as new paragraph has been added in the Results and Discussion section.
Changes in Experimental section with the incorporation of “X-Ray diffraction analysis”
“XRD analysis of the TiO2 NPs have been made using a BRUKER D8 Discover diffractometer fitted with a copper source (Kα=1.5406 Å) under Bragg Brentano configuration and 2q scanning between 20º and 80º with 0.02º and 1 s per step. A 0.1mm nickel sheet was placed just before the detector in order to eliminate the Cu- Kb radiation on the diffractograms”.
Changes on the “Results and Discussion” section
“As it has been previously commented in Scheme 1, the incorporation of TiO2 NPs into the sol-gel hybrid matrix has been performed with the aim of improving the corrosion resistance of the coatings. First of all, in order to clarify the nature of these TiO2 NPs into the sol-gel coatings, a XRD analysis of these TiO2 NPs have been made as it can be appreciated in Figure 1. The XRD diffractogram indicates that the resultant titanium oxide used for the fabrication of the sol-gel hybrid coatings consists of a mixture of rutile and anatase because the presence of both phases can be clearly observed in the XRD diffractogram”.
“Figure 1: XRD diffractogram of the TiO2 NPs used for the fabrication of the sol-gel coatings which is composed of a mixture of rutile and anatase forms”.
3) I also think that for improving the quality of the paper, the correlation of the observed wettability properties should be better addressed, for example, with the top morphology (acquired by higher resolution SEM images of the Top layer / AFM images or surface chemistry properties (i.e. IR taken on the surface).
We would like to thank to the reviewer for the opportunity to enlighten this key point which is of vital importance for a better understanding of the manuscript. New SEM and AFM images (Figures 9 and 10 in the revised version) have been incorporated in order to address the existence of morphological evidences between the observed wettability properties and the resultant surface morphology. Finally, new references has been also incorporated related to the use of the PFAS precursor and its resultant hydrophobic behavior derived from the combination of both low surface energy and enhanced microscale roughness.
Changes on the “Characterization of the sol-gel hybrid coatings” section
“In addition, the topography along with the surface has been also examined by using atomic force microscopy (AFM) in tapping mode (Veeco Innova AFM, Veeco Instruments) for a scan area of 20 x 20 μm”.
Changes on the “Results and Discussion” section
“In order to have a better characterization of the resultant morphology of the sol-gel hybrid coatings, SEM and AFM analysis were performed. As it can be previously commented, in Figure 9 it is presented the SEM images of the topographic surface which clearly indicate a microporous morphology throughout the surface of the aluminum substrate because it can be appreciated the presence of variable microgrooves. These microgrooves have the tendency to entrap air in the pores of the films which contributes to the beading up and easy rolling of the water droplets over the protrusions, making possible an enhancement in the water repellency with a higher water contact angle value) [44, 62, 63]. The use of PFAS in the outer surface makes possible an increase in the surface roughness (corroborated by SEM images) which is one of the most important requirements to achieve hydrophobic character along with the low interfacial energy. According to this, the surface energy of functional groups is decreased in the following order –CF3 < –CF2H < –CF2 < –CH3 < –CH2, hence the use of the PFAS precursor in the outer surface which is composed of fluorinated functional groups (CF3 and CF2) which enables a considerable enhancement in the hydrophobic and water repellent properties [64, 65]”.
“Figure 10: SEM images of the topographic surface of the sol-gel hybrid coatings for a scale bar of 50 µm (a) and 30 µm (b), respectively”.
“In addition, AFM analysis was also performed for the characterization of the topographic surface of the sol-gel hybrid coating. In Figure 10, it is shown two dimension (2D) height image (Fig. 10a), three dimension (3D) topographic surface (Fig. 10b) and the section profile in three different regions of the surface (Fig. 10c) for a scan area of 20 x 20 μm, respectively. From the AFM images of Fig. 10a and 10b can be observed a microscale roughness, whereas from the profile section (Fig. 10c) can be appreciated a variable number of protrusions (hills) along the topographic surface of three different regions of analysis. These morphological evidences associated to the terminal perfluoro groups in the outer surface of the sol-gel coatings allows an entrapment of enough air to prevent the penetration of water into the protrusions, making possible a reduction of the contact area of the water droplets with the surface and as a result, an improvement of the water repellency behavior is obtained [43, 44, 66]. To sum up, the use of this PFAS precursor for the fabrication of the hybrid sol-gel coatings in the outer part of the coating makes possible a dual synergetic effect such as a considerable enhancement of the surface roughness of the coating as well as a lowering of the surface energy, respectively”.
“Figure 11: AFM images in tapping mode of the resultant topographic surface of the sol-gel hybrid coatings in 2D (a), 3D (b) and the resultant section profiles in three different locations of the coatings, respectively”.
Changes on the “Bibliography” section
The following references related to the behavior of PFAS in the outer surface of the sol-gel coatings have been also incorporated in the new version of the manuscript in order to address the morphological evidences by using this specific precursor with the resultant hydrophobic behavior derived from the combination of both low surface energy and enhanced microscale roughness.
(62) Sermon, P. A.; Leadley, J. G. Fluoroalkylsilane Modification of Sol-Gel SiO2-TiO2 Coatings. J. Sol Gel Sci. Technol. 2004, 32, 293-296.
(63) Liu, W.; Luo, Y.; Sun, L.; Wu, R.; Jiang, H.; Liu, Y. Fabrication of the Superhydrophobic Surface on Aluminum Alloy by Anodizing and Polymeric Coating. Appl. Surf. Sci. 2013, 264, 872-878.
(64) Rivero, P.; Yurrita, D.; Berlanga, C.; Palacio, Jose F.; Rodriguez, R. Functionalized electrospun fibers for the design of novel hydrophobic and anticorrosive surfaces. Coatings, 2018, 8(9), 300.
(65) Liang, J.; Wang, L.; Bao, J.; He, L. SiO2-g-PS/fluoroalkylsilane Composites for Superhydrophobic and Highly Oleophobic Coatings. Colloids Surf. A Physicochem. Eng. Asp. 2016, 507, 26-35.
(66) Guo, Z.; Zhou, F.; Hao, J.; Liu, W. Effects of System Parameters on Making Aluminum Alloy Lotus. J. Colloid Interface Sci. 2006, 303, 298-305.
4) With particular reference to coatings with TiO2 nanoparticle and the related wettability property, the authors give their main focus to protection corrosion resistance issues, but there are also some other applications where such type of coating have been addressed, including photocatalysis, bacteriostatic activity, etc. I think relevant to include a general comment about these kinds of properties. The authors can refer to: Cravanzola, S., RSC Advances 5, 2015, 103255; Liu J., RSC Advances 7, 2017, 43938; De Falco G., Surface and Coatings Technology 349, 2018, 830(2015).
We totally agree the reviewer and we would like to thank to the reviewer for the opportunity to enlighten this important aspect related to the use of the TiO2 nanoparticles in other different fields such as photocatalysis, antibacterial activity, UV blocking function or even self-cleaning properties, among others. New references have been incorporated in the revised manuscript in order to address the multifunctionality of this type of nanoparticles and a new paragraph has been incorporated in the Introduction section.
Changes on the “Introduction” section
“These nanoparticles are prepared in different forms (anatase or rutile forms), showing unique properties such as non-toxicity, excellent antibacterial activity, good compatibility with various materials, high chemical stability at high temperatures, high photocatalytic activity, photo-stability and ability of absorbing ultraviolet light. A representative example of this great multifunctionality related to TiO2 NPs can be found in [54] where its specific mechanism action is clearly explained, whereas potential applications of titanium dioxide materials can be found in [55]. According to this, the use of these nanoparticles makes possible the fabrication of advanced coatings with improved corrosion resistance and antibacterial properties due to its intrinsic excellent antimicrobial properties [56, 57]. Moreover, other research field of the TiO2-based materials can be found in environmental and energy related applications such as photocatalysis and photovoltaics [58, 59]”.
Changes on the “Bibliography” section
The following references have been added in the revised manuscript:
(54) Radetic, M. Functionalization of Textile Materials with TiO2 nanoparticles. J. Photochem. Photobiol. C Photochem. Rev. 2013, 16, 62-76.
(55) Chen, X.; Mao, S. S. Titanium Dioxide Nanomaterials: Synthesis, Properties, Modifications and Applications. Chem. Rev. 2007, 107, 2891-2959.
(56) Liu, J.; Lou, Y.; Zhang, C.; Yin, S.; Li, H.; Sun, D.; Sun, X. Improved Corrosion Resistance and Antibacterial Properties of Composite Arch-Wires by N-Doped TiO2coating. RSC Adv. 2017, 7, 43938-43949.
(57) De Falco, G.; Ciardiello, R.; Commodo, M.; Del Gaudio, P.; Minutolo, P.; Porta, A.; D'Anna, A. TIO2 Nanoparticle Coatings with Advanced Antibacterial and Hydrophilic Properties Prepared by Flame Aerosol Synthesis and Thermophoretic Deposition. Surf. Coat. Technol. 2018, 349, 830-837.
(58) Cravanzola, S.; Jain, S. M.; Cesano, F.; Damin, A.; Scarano, D. Development of a Multifunctional TiO2/MWCNT Hybrid Composite Grafted on a Stainless Steel Grating. RSC Adv. 2015, 5, 103255-103264.
(59) Bell, N. J.; Ng, Y. H.; Du, A.; Coster, H.; Smith, S. C.; Amal, R. Understanding the Enhancement in Photoelectrochemical Properties of Photocatalytically Prepared TiO2-Reduced Graphene Oxide Composite. J. Phys. Chem. C 2011, 115, 6004-6009.
5) Grammar/English:
All the following grammatical mistakes have been corrected in the revised version of the manuscript. All these changes have been highlighted in yellow for a better localization of them. In addition, other different grammatical errors have been also corrected in this revised version and new paragraphs have been rewritten for a better understanding of the manuscript. Finally, we would like to thank to the anonymous reviewer for his/her positive comments and we hope that this final version can be published in Metals.
Please, correct.
“In addition, the simplicity, versatility and easiness of the fabrication process by using the dip-coating technique makes possible the development of effective …” (line 97, page 3). Please, correct the verb.
The verb has been corrected. In the revised version appears “allows” instead of “makes”.
“This combination of uniform distributed TiO2 …” (line 220, page 6). Please substitute “uniform” with an adverb.
The word “uniform” has been substituted by “uniformly” in the revised version.
“As it can be observed in Table 3, the pitting corrosion potential for different sol-gel samples are analysed …” (line 276, page 8). Please, correct the verb.
The verb has been corrected. In the revised version appears “is analysed” instead of “are analysed”.
“In addition, the presence of other specific chemical elements are also observed …” (line 331, page 11). Please, correct the verb.
The verb has been corrected. In the revised version appears “is also observed” instead of “are also observed”.
“Finally, the corrosion behaviour of the sol-gel hybrid coatings have been also analyzed” (line 351, page 12). Please, correct the verb.
The verb has been corrected. In the revised version appears “has been also analyzed” instead of “have been also analyzed”.
“It has been demonstrated that the presence of these TiO2NPs produce a surface modification” (line 384, page13). Please, correct the verb.
The verb has been corrected. In the revised version appears “produces” instead of “produce”.
“In addition, the importance of a further thermal treatment (curing step) is also studied because a considerable enhancement of the mechanical properties have been obtained” (line 389, Page 13). Please rewrite the sentence and correct the verb.
According to the demand from the reviewer, this sentence has been rewritten for a better understanding and the verb has been also changed, as it can be observed in the revised manuscript.
Changes on the “Conclusions”
“In addition, the pencil hardness test has shown a considerable enhancement of the mechanical characteristics due to the thermal post-treatment (curing step)”.

Reviewer 2 Report
The authors devise a coating system for aluminum that improves the durability of the alloy. The authors give a good review of the literature since 2000.
In order to prepare the coatings, there are many steps. The authors describe the steps in detail. If anyone wants to repeat the process, there is enough detail to make this possible. The worry is that there are so many steps that the process may never be scalable. How can the coatings become practical? What would it take to implement this coating system?
The authors perform appropriate tests to determine the corrosion protection ability of the coatings. The results and discussion are thorough.
The heat treatments in Tables 2 and 3 did not print properly on my copy of the paper. The authors need to check the data in the Tables.
The paper is acceptable after careful proofreading and copy-editing. An English editor or someone with excellent English skills should look over this. For example, the preferred spelling is “alkoxide” instead of “alkoxyde”.
Author Response
Comments from Reviewer_2:
The authors devise a coating system for aluminum that improves the durability of the alloy. The authors give a good review of the literature since 2000.
First of all, we would like to thank to the anonymous reviewer for his/her positive criticisms. We hope that this revised version can be published in Metals.
All the changes have been highlighted in yellow for a better understanding and localization of them, as it can be appreciated in the revised manuscript. Finally, a native English speaking colleague with excellent skills in thin-films fabrication has thoroughly revised the manuscript.
1. In order to prepare the coatings, there are many steps. The authors describe the steps in detail. If anyone wants to repeat the process, there is enough detail to make this possible. The worry is that there are so many steps that the process may never be scalable. How can the coatings become practical? What would it take to implement this coating system?
We would like to thank to the reviewer for this interesting comment. We totally agree that the fabrication process is based on numerous steps for obtaining a dual synergetic effect of anticorrosion and hydrophobicity. These coatings can become practical if the deposition technique is based on spray coating or even roll-to-roll technology instead of dip-coating process, as it is proposed in this work. The use of these alternative techniques by using sol-gel precursors can be a great alternative of implementation in global coatings market. According to this issue related to the scalability and its corresponding implementation, a new paragraph has been incorporated in the revised version of the manuscript.
Changes on the “Introduction” section
“Scalability must be always a main concern in the choice of solutions for these new problems. Sol-gel coating technology is susceptible of being scaled by implementing spray coating methods or, in particular cases, roll-to-roll processes. In addition, sol-gel technology is a potential candidate for corrosion protection because it shows interesting features such as a good barrier effect, possibility of incorporating inhibitors into the matrix, good adhesion over different metallic substrates and good compatibility with additional top layers”.
2. The authors perform appropriate tests to determine the corrosion protection ability of the coatings. The results and discussion are thorough.
We would like to thank to the reviewer for this positive comme
nt. We totally agree that the results clearly indicate that the effect of combining TiO2NPs in the inner part of the coating with fluorinated polymeric chains derived from PFAS precursor in the outer part of the coating has produced an enhancement in the corrosion protection, as it has been corroborated by the corrosion tests.
3. The heat treatments in Tables 2 and 3 did not print properly on my copy of the paper. The authors need to check the data in the Tables.
Thank you very much for this important suggestion. In the new revised version of the manuscript, the symbols of heat treatment in Tables 2 and 3 have been changed by the words “yes/no” in this specific column of the Tables in order to indicate if the samples have been thermally treated or no, respectively.
4. The paper is acceptable after careful proofreading and copy-editing. An English editor or someone with excellent English skills should look over this. For example, the preferred spelling is “alkoxide” instead of “alkoxyde”.
According to the demand from the reviewer, the words “alkoxyde” have been replaced by “alkoxide” in the revised version of the manuscript.

Reviewer 3 Report
Although basic idea of good anti-corrosion effect by TiO2 and fluorinated chains are well expectable and might not be so novel, this work provides precious practical data. Publishing these data in public media is meaningful. From this positive viewpoint, I recommend publication of this work in Metals. However, it looks somehow old-fashioned work. Therefore, some revisions are necessary to make it more attractive.
1) Reference selection is rather old and description of Introduction is somehow old-fashioned. These factors may make this work less attractive. Because aluminum coating is actually important in some advanced technology such as modern batteries, short descriptions on such wide range impact and importance of aluminum coatings had better be added to Introduction with citing very new paper (for example, see, Bull. Chem. Soc. Jpn. 2018, 91, 1660-1682). This modification make this work looking more attractive.
2) In order to clearly show superiority and advantages of the current methods, comparisons over the past typical results on anti-corrosion coating in literatures have to be done quantitatively. Making one table for such comparison through literature search would be a good way.
3) This work does not include many morphological evidences. Figure 7 provides some SEM images. However, they were not clearly interpreted. Top image shows many holes (Figure 7a), but cross-sectional image looks very smooth. Hole depth (where they reach at surface or not) is important to explain anti-corrosion capability. Please make much deeper discussions on such morphological aspects.
Author Response
Comments from Reviewer_3:
Although basic idea of good anti-corrosion effect by TiO2 and fluorinated chains are well expectable and might not be so novel, this work provides precious practical data. Publishing these data in public media is meaningful. From this positive viewpoint, I recommend publication of this work in Metals. However, it looks somehow old-fashioned work. Therefore, some revisions are necessary to make it more attractive.
First of all, we would like to thank to the anonymous reviewer for his/her positive criticisms. We hope that this revised version can be published in Metals.
All the changes have been highlighted in yellow for a better understanding and localization of them, as it can be appreciated in the revised manuscript. Finally, a native English speaking colleague with excellent skills in thin-films fabrication has thoroughly revised the manuscript.
1) Reference selection is rather old and description of Introduction is somehow old-fashioned. These factors may make this work less attractive. Because aluminum coating is actually important in some advanced technology such as modern batteries, short descriptions on such wide range impact and importance of aluminum coatings had better be added to Introduction with citing very new paper (for example, see, Bull. Chem. Soc. Jpn. 2018, 91, 1660-1682). This modification makes this work looking more attractive.
We would like to thank to the reviewer for this positive comment. We totally agree that the Introduction section has to be modified, and due to this, new paragraphs have been added in the revised manuscript with the aim to make this work more attractive for the reader. In this sense, the use of aluminum in the development of new technologies for energy production and storage (batteries or capacitors) is commented. Finally, the bibliography section has been also updated with a more recent and actual list of references.
Changes on the “Introduction” section
“Aluminum and its alloys are, after steel and cast irons, the group of metallic materials most widely used by the industry. Their unique combination of low density, mechanical properties, high thermal and electrical conductivity and affordable cost make aluminum the choice material for an increasing number of applications such as airframes, automotive components, power lines, heat exchangers or food containers”.
“Despite of the low strength and high ductility of the pure metal, many of its alloys such as those of the 2XXX, 6XXX and 7XXX series can reach excellent mechanical properties through solution and ageing heat treatments [1]. The AA6061 is a paradigmatic example, being one of the most employed alloys in aeronautics as well as in the automobile industry or in shipbuilding. In addition to the above mentioned applications, aluminum is also playing a major role in the development of new technologies for energy production and storage. Bipolar plates employed in PEM fuel cells are perhaps the best studied example of aluminum based components working in a chemically aggressive environment [2, 3]. On the other hand, among energy storage systems where aluminum electrodes are a key element it is worth mentioning capacitors [4] and new batteries [5-7]”.
“For structural applications, in environments where chloride ions can be present, the use of chromates as corrosion inhibitors has been the standard solution for providing the required protection in aqueous media. In the case of aluminum alloys this can be implemented chromic acid anodized and chromate conversion coatings [10, 11]. However, in the last two decades, the use of hexavalent chromium compounds has been banned in most industrialized countries because they are extremely dangerous for the ecosystem and human health [12, 13]. Due to these serious drawbacks, the scientific community has been looking for safer and ecofriendly alternatives with the aim of replacing these toxic chromium compounds [14]. In the case of AA6061 bipolar plates, PANI and PPy films [2] as well as PVD coatings for the AA5083 [15] have been suggested as a possible solution for preventing corrosion in PEM fuel cells”.
“Scalability must be always a main concern in the choice of solutions for these new problems. Sol-gel coating technology is susceptible of being scaled by implementing spray coating methods or, in particular cases, roll-to-roll processes”.
Changes on the “Bibliography” section
The following references have been added in the revised manuscript:
(2) Antunes, R. A.; Oliveira, M. C. L.; Ett, G.; Ett, V. Corrosion of Metal Bipolar Plates for PEM Fuel Cells: A Review. Int J Hydrogen Energy 2010, 35, 3632-3647.
(3) Deyab, M. A. Corrosion Protection of Aluminum Bipolar Plates with Polyaniline Coating Containing Carbon Nanotubes in Acidic Medium Inside the Polymer Electrolyte Membrane Fuel Cell. J. Power Sources 2014, 268, 50-55.
(4) Fan, Z.; Zhu, C.; Chang, F. Effect of Thiourea on the Corrosion Behavior of Aluminum Anode Foil of Aluminum Electrolytic Capacitor. J Mater Eng Perform 2018, 27, 4168-4175.
(5) Watanabe, M.; Dokko, K.; Ueno, K.; Thomas, M. L. From Ionic Liquids to Solvate Ionic Liquids: Challenges and Opportunities for Next Generation Battery Electrolytes. Bull. Chem. Soc. Jpn. 2018, 91, 1660-1682.
(6) Yamada, Y.; Chiang, C. H.; Sodeyama, K.; Wang, J.; Tateyama, Y.; Yamada, A. Corrosion Prevention Mechanism of Aluminum Metal in Superconcentrated Electrolytes. ChemElectroChem 2015, 2, 1687-1694.
(7) Wang, R.; Li, W.; Liu, L.; Qian, Y.; Liu, F.; Chen, M.; Guo, Y.; Liu, L. Carbon black/graphene-Modified Aluminum Foil Cathode Current Collectors for Lithium Ion Batteries with Enhanced Electrochemical Performances. J Electroanal Chem 2019, 833, 63-69.
(15) Barranco, J.; Barreras, F.; Lozano, A.; Lopez, A. M.; Roda, V.; Martin, J.; Maza, M.; Fuentes, G. G.; Almandoz, E. Cr and Zr/Cr Nitride CAE-PVD Coated Aluminum Bipolar Plates for Polymer Electrolyte Membrane Fuel Cells. Int J Hydrogen Energy 2010, 35, 11489-11498.
2) In order to clearly show superiority and advantages of the current methods, comparisons over the past typical results on anti-corrosion coating in literatures have to be done quantitatively. Making one table for such comparison through literature search would be a good way.
We totally agree that a comparative table about the advantages of the current methods based on the sol-gel technology for the development of coatings for corrosion protection has to be added in the revised version of the manuscript. First of all, it is important to remind that chromate conversion coatings are banned in most industrialized countries for being carcinogenic and due to this, the environmental regulations require the development of environmentally safe alternatives. Among all them, the sol-gel technology is a potential candidate because it shows interesting features such as a good barrier effect, possibility of incorporating inhibitors into the matrix, good adhesion to the metallic substrate and good compatibility with additional top layers.
Changes on the “Introduction” section
“Scalability must be always a main concern in the choice of solutions for these new problems. Sol-gel coating technology is susceptible of being scaled by implementing spray coating methods or, in particular cases, roll-to-roll processes. In addition, sol-gel technology is a potential candidate for corrosion protection because it shows interesting features such as a good barrier effect, possibility of incorporating inhibitors into the matrix, good adhesion over different metallic substrates and good compatibility with additional top layers”.
“Finally, a summary of some of different sol-gel alternatives previously commented based on the design of effective coatings for corrosion protection can be appreciated in Table 1 as a function of the selected metallic substrate, precursors used for the fabrication of the sol-gel coating (simple or hybrid matrix) and presence or absence of active agent.
“Table 1. Summary of the fabrication of different sol-gel coatings as a function of the selected substrate, precursors and the presence orabsence of active agents”.
Metallic substrate | Precursors for the fabrication of the sol-gel coating | Active agent | Ref. |
AA2024-T3 | Tetramethylortosilicate | - | [17] |
Carbon steel | Zirconium tetrabutoxide | - | [18] |
Mild steel | -glycidoxypropyltrimethoxysilane (GPTMS) and aminopropylethoxysilane | - | [20] |
AA5754 | Tetraethylorthosilicate (TEOS) | - | [25] |
AISI 304 | Tetraethylorthosilicate (TEOS) | - | [27] |
AISI 304 | Tetraethylorthosilicate (TEOS) and 3-methacryloxypropyltrimethoxysilane (MPS) | - | [28] |
AA2024-T3 | 3-glycidoxypropyltrimethoxysilane (GPTMS) and titanium organic compounds | - | [32] |
AA2024-T3 | Tetramethoxysilane (TMOS) and 3-glycidoxypropyltrimethoxysilane (GPTMS) | Organic corrosion inhibitors | [34] |
AA2024-T3 | Vinyltrimethoxysilane (VTMS) and tetraethylorthosilicate (TEOS) | Ethylenediamine tetra (methylene phosponic acid) | [35] |
AA3005 | Tetraethylorthosilicate (TEOS) and methyltriethoxysilane (MTES) | Cerium salts | [36] |
AA2024-T3 | Tetramethoxysilane (TMOS) and 3-glycidoxypropyltrimethoxysilane (GPTMS) | Organic corrosion inhibitors | [39] |
AA6061T6 | Methyltriethoxysilane (MTEOS), 3-glycidoxypropyltrimethoxysilane (GPTMS) and perfluoroalkylsilane | Graphene oxide | [43] |
AA2024 | Tetraethylorthosilicate (TEOS) and 3-methoxysilylpropylmethacrylate (TSPM) | TiO2-CeO2 nanoparticles | [50] |
3) This work does not include many morphological evidences. Figure 7 provides some SEM images. However, they were not clearly interpreted. Top image shows many holes (Figure 7a), but cross-sectional image looks very smooth. Hole depth (where they reach at surface or not) is important to explain anti-corrosion capability. Please make much deeper discussions on such morphological aspects.
We would like to thank to the reviewer for the opportunity to enlighten this key point which is of vital importance for a better understanding of the manuscript. First of all, new SEM and AFM images have been incorporated in the revised version in order to address morphological evidences between the observed wettability properties and the surface morphology.
As it can be appreciated in the new SEM images, the topographic surface of the sol-gel hybrid coatings clearly shows a microporous morphology throughout the surface of the aluminum substrate because it can be appreciated the presence of microgrooves in the outer surface. These microgrooves have the tendency to entrap air in the pores of the films which contributes to the beading up and easy rolling of the water droplets over the protrusions, making possible an enhancement in the water repellency (with a higher water contact angle value) as well as in the corrosion resistance (demonstrated by the corrosion tests).
Figure 1: SEM images of the topographic surface of the sol-gel hybrid for a scale bar of 50 µm (a) and 30 µm (b), respectively.
In addition, this enhancement in the resultant wettability is associated to the use of the PFAS precursor for the fabrication of the sol-gel hybrid coatings because this precursor makes possible an increase in the surface roughness (previously observed in SEM images) which is one of the most important requirements to achieve hydrophobic character along with the low interfacial energy. According to this, the surface energy of functional groups is decreased in the following order –CF3 < –CF2H < –CF2 < –CH3 < –CH2, hence the use of the PFAS precursor in the outer surface which is composed of fluorinated functional groups (CF3 and CF2) which enables a considerable enhancement in the hydrophobic and water repellent properties.
Finally, AFM analysis has been also performed in order to have a better characterization of the topographic surface of the sol-gel hybrid coating. In the following figure, it is shown two dimension (2D) height image (a), three dimension (3D) topographic surface image (b) and the section profile in three different regions of the surface (c) for a scan area of 20 x 20 μm, respectively. From these AFM images can be clearly observed a combination of microscale and nanoscale roughness, whereas from the profile section can be appreciated a variable number of protrusions (hills) along the topographic surface of the three different regions of analysis. These morphological evidences associated to the terminal perfluoro groups in the outer surface of the sol-gel coatings allows an entrapment of enough air to prevent the penetration of water into the protrusions, making possible a reduction of the contact area of the water droplets with the surface and as a result, an improvement of the water repellency behavior is obtained. To sum up, the use of this PFAS precursor for the fabrication of the hybrid sol-gel coatings in the outer part of the coating makes possible a dual synergetic effect such as a considerable enhancement of the surface roughness of the coating as well as a lowering of the surface energy, respectively.
Figure 2: AFM images in tapping mode of the resultant topographic surface of the sol-gel hybrid coatings in 2D (a), 3D (b) and the resultant section profiles in three different locations of the coatings, respectively.
Changes on the “Characterization of the sol-gel hybrid coatings” section
“In addition, the topography along with the surface has been also examined by using atomic force microscopy (AFM) in tapping mode (Veeco Innova AFM, Veeco Instruments) for a scan area of 20 x 20 μm”.
Changes on the “Results and Discussion” section
“In order to have a better characterization of the resultant morphology of the sol-gel hybrid coatings, SEM and AFM analysis were performed. As it can be previously commented, in Figure 9 it is presented the SEM images of the topographic surface which clearly indicate a microporous morphology throughout the surface of the aluminum substrate because it can be appreciated the presence of variable microgrooves. These microgrooves have the tendency to entrap air in the pores of the films which contributes to the beading up and easy rolling of the water droplets over the protrusions, making possible an enhancement in the water repellency with a higher water contact angle value) [44, 62, 63]. The use of PFAS in the outer surface makes possible an increase in the surface roughness (corroborated by SEM images) which is one of the most important requirements to achieve hydrophobic character along with the low interfacial energy. According to this, the surface energy of functional groups is decreased in the following order –CF3 < –CF2H < –CF2 < –CH3 < –CH2, hence the use of the PFAS precursor in the outer surface which is composed of fluorinated functional groups (CF3 and CF2) which enables a considerable enhancement in the hydrophobic and water repellent properties [64, 65]”.
“Figure 10: SEM images of the topographic surface of the sol-gel hybrid coatings for a scale bar of 50 µm (a) and 30 µm (b), respectively”.
“In addition, AFM analysis was also performed for the characterization of the topographic surface of the sol-gel hybrid coating. In Figure 10, it is shown two dimension (2D) height image (Fig. 10a), three dimension (3D) topographic surface (Fig. 10b) and the section profile in three different regions of the surface (Fig. 10c) for a scan area of 20 x 20 μm, respectively. From the AFM images of Fig. 10a and 10b can be observed a microscale roughness, whereas from the profile section (Fig. 10c) can be appreciated a variable number of protrusions (hills) along the topographic surface of three different regions of analysis. These morphological evidences associated to the terminal perfluoro groups in the outer surface of the sol-gel coatings allows an entrapment of enough air to prevent the penetration of water into the protrusions, making possible a reduction of the contact area of the water droplets with the surface and as a result, an improvement of the water repellency behavior is obtained [43, 44, 66]. To sum up, the use of this PFAS precursor for the fabrication of the hybrid sol-gel coatings in the outer part of the coating makes possible a dual synergetic effect such as a considerable enhancement of the surface roughness of the coating as well as a lowering of the surface energy, respectively”.
“Figure 11: AFM images in tapping mode of the resultant topographic surface of the sol-gel hybrid coatings in 2D (a), 3D (b) and the resultant section profiles in three different locations of the coatings, respectively”.
Changes on the “Bibliography” section
The following references related to the behavior of PFAS in the outer surface of the sol-gel coatings have been also incorporated in the new version of the manuscript in order to address the morphological evidences by using this specific precursor with the resultant hydrophobic behavior derived from the combination of both low surface energy and enhanced microscale roughness.
(62) Sermon, P. A.; Leadley, J. G. Fluoroalkylsilane Modification of Sol-Gel SiO2-TiO2 Coatings. J. Sol Gel Sci. Technol. 2004, 32, 293-296.
(63) Liu, W.; Luo, Y.; Sun, L.; Wu, R.; Jiang, H.; Liu, Y. Fabrication of the Superhydrophobic Surface on Aluminum Alloy by Anodizing and Polymeric Coating. Appl. Surf. Sci. 2013, 264, 872-878.
(64) Rivero, P.; Yurrita, D.; Berlanga, C.; Palacio, Jose F.; Rodriguez, R. Functionalized electrospun fibers for the design of novel hydrophobic and anticorrosive surfaces. Coatings, 2018, 8(9), 300.
(65) Liang, J.; Wang, L.; Bao, J.; He, L. SiO2-g-PS/fluoroalkylsilane Composites for Superhydrophobic and Highly Oleophobic Coatings. Colloids Surf. A Physicochem. Eng. Asp. 2016, 507, 26-35.
(66) Guo, Z.; Zhou, F.; Hao, J.; Liu, W. Effects of System Parameters on Making Aluminum Alloy Lotus. J. Colloid Interface Sci. 2006, 303, 298-305.

Round 2
Reviewer 1 Report
I believe that the paper has been significantly improved by the authors and can be accepted for publication in the present form.
Reviewer 2 Report
Revisions are acceptable.
Reviewer 3 Report
Revisions are fine. The revised version becomes acceptable.